# Universal Beta Splatting

**Rong Liu**[1,2,*], **Zhongpai Gao**[2,†], **Benjamin Planche**[2], **Meida Chen**[1],
**Van Nguyen Nguyen**[2], **Meng Zheng**[2], **Anwesa Choudhuri**[2],
**Terrence Chen**[2], **Yue Wang**[1], **Andrew Feng**[1], **Ziyan Wu**[2]

[1]University of Southern California, Los Angeles, CA, USA
[2]United Imaging Intelligence, Boston, MA, USA

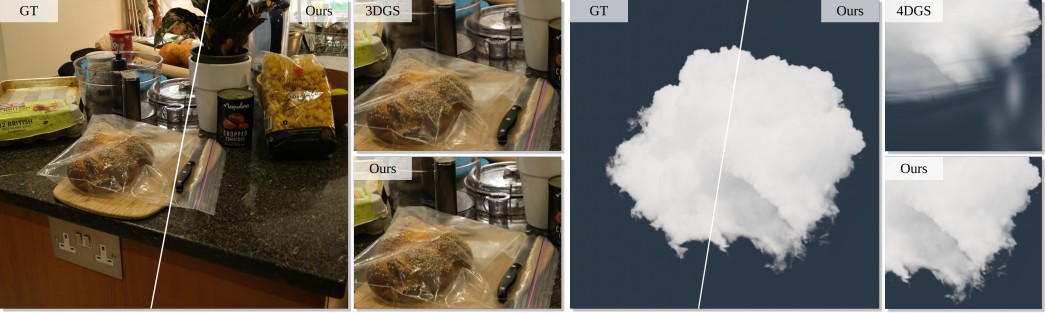

Figure 1: Visualization of UBS rendering quality. For static real-world scenes (left), UBS achieves superior rendering of reflective and specular materials compared to 3DGS (Kerbl et al., 2023). For dynamic volumetric scenes (right), UBS maintains high visual fidelity in complex spatio-temporal scenarios where 4DGS (Yang et al., 2024a) produces blurring artifacts.

## Abstract

We introduce Universal Beta Splatting (UBS), a unified framework that generalizes 3D Gaussian Splatting to N-dimensional anisotropic Beta kernels for explicit radiance field rendering. Unlike fixed Gaussian primitives, Beta kernels enable controllable dependency modeling across spatial, angular, and temporal dimensions within a single representation. Our unified approach captures complex light transport effects, handles anisotropic view-dependent appearance, and models scene dynamics without requiring auxiliary networks or specific color encodings. UBS maintains backward compatibility by approximating to Gaussian Splatting as a special case, guaranteeing plug-in usability and lower performance bounds. The learned Beta parameters naturally decompose scene properties without explicit supervision: spatial (surface vs. texture), angular (diffuse vs. specular), and temporal (static vs. dynamic). Our CUDA-accelerated implementation achieves real-time rendering while consistently outperforming existing methods across static, view-dependent, and dynamic benchmarks, establishing Beta kernels as a scalable universal primitive for radiance field rendering. Our project website is available at https://rongliu-leo.github.io/universal-beta-splatting/.

## 1 Introduction

Real-time photorealistic rendering of complex scenes remains a fundamental challenge in computer vision and graphics. While Neural Radiance Fields (NeRF) (Mildenhall et al., 2020) achieve exceptional visual quality through continuous volumetric representations, their reliance on dense ray sampling imposes prohibitive computational costs. 3D Gaussian Splatting (3DGS) (Kerbl et al., 2023)

---

*This work was done during Rong Liu's internship at United Imaging Intelligence, Boston, MA, USA.
†Corresponding author.

addresses efficiency through explicit primitive-based rendering, achieving real-time performance with competitive fidelity. However, Gaussian kernels limit representational capacity: their fixed bell-shaped profiles struggle with sharp boundaries, while view-dependent effects require auxiliary spherical harmonic encodings that fragment the representation. Dynamic scene extensions (Yang et al., 2024b; Luiten et al., 2024) further increase complexity through additional deformation networks.

Recent advances partially address these limitations through specialized designs. Deformable Beta Splatting (DBS) (Liu et al., 2025a) improves geometric fidelity using spatially adaptive Beta kernels, but remains limited to 3D spatial dimensions and still requires separate Spherical Beta functions for view-dependent color. $N$-dimensional Gaussian extensions like 6DGS (Gao et al., 2025a) and 7DGS (Gao et al., 2025b) incorporate view and temporal dimensions through conditional distributions, enabling them to capture scattering effects and dynamic changes. However, these approaches remain constrained by the Gaussian kernel's symmetric profile across dimensions—preventing independent control of spatial sharpness, angular specularity, and temporal dynamics.

The core insight is that different scene properties require different kernel behaviors. Spatial geometry benefits from adaptive sharpness—flat for surfaces, peaked for textures. Angular appearance spans from diffuse to specular responses. Temporal dynamics range from static (constant support) to rapid motion (localized activation). While Gaussian kernels couple all dimensions with identical bell-shaped profiles, real scenes exhibit independent variations across space, angle, and time.

We introduce Universal Beta Splatting (UBS), a unified framework that generalizes radiance field rendering to $N$-dimensional anisotropic Beta kernels. Unlike fixed Gaussian profiles, Beta kernels provide per-dimension shape control through learnable parameters, enabling each dimension to adopt its optimal form. This allows a single primitive type to simultaneously model spatial geometry, view-dependent appearance, and temporal dynamics while maintaining computational efficiency.

Our approach addresses three technical challenges: (1) spatial-orthogonal Cholesky parameterization preserving orthonormal spatial structure while enabling flexible cross-dimensional correlations, (2) Beta-modulated conditional slicing transforming high-dimensional primitives into renderable 3D representations with dimension-specific anisotropy, and (3) fully accelerated CUDA implementation ensuring real-time performance.

Crucially, UBS maintains backward compatibility: when Beta parameters equal zero, the kernel approximates to Gaussian, recovering 3DGS, 6DGS, or 7DGS depending on dimensionality. This ensures seamless drop-in replacement with guaranteed performance lower bounds. Beyond rendering improvements, learned Beta parameters naturally decompose scenes into interpretable components without explicit supervision: spatial parameters separate geometry from textures, angular parameters distinguish diffuse from specular, and temporal parameters isolate static from dynamic elements.

We validate UBS across static, view-dependent, and dynamic benchmarks, achieving state-of-the-art visual quality—improving PSNR by up to +8.27 dB on static scenes and +2.78 dB on dynamic scenes compared to corresponding Gaussian baselines. Our contributions are:

- **Universal Beta Splatting:** A unified $N$-dimensional representation with per-dimension shape control, enabling simultaneous modeling of spatial, angular, and temporal properties through anisotropic Beta kernels with spatial-orthogonal Cholesky parameterization.
- **Efficient CUDA implementation:** Fully accelerated differentiable rendering with custom kernels achieving real-time performance.
- **Interpretable scene decomposition:** Emergent separation of geometric, appearance, and motion components through learned Beta parameters without supervision.
- **Backward compatibility:** Reduction to approximate existing methods as special cases, ensuring plug-in usability with performance lower bounds while enabling substantial improvements.

## 2 RELATED WORK

**Neural Radiance Fields and Gaussian Splatting.** Neural Radiance Fields (NeRF) (Mildenhall et al., 2020) revolutionized novel view synthesis through continuous volumetric representations. While extensions like Instant-NGP (Müller et al., 2022), Mip-NeRF (Barron et al., 2022), and Zip-NeRF (Barron et al., 2023) improved training speed and quality, NeRF methods remain limited by slow inference and inability to model off-ray transport phenomena.

3D Gaussian Splatting (3DGS) (Kerbl et al., 2023) achieved real-time rendering through explicit anisotropic Gaussian primitives. Extensions address anti-aliasing (Yu et al., 2024; Yan et al., 2024), compression (Lee et al., 2024a; Morgenstern et al., 2024), and semantic applications (Shi et al., 2024; Zhou et al., 2024; Peng et al., 2025; Xiong et al., 2026). However, 3DGS's spherical harmonic encoding struggles with complex view-dependent effects. To better capture view dependence, $N$-dimensional Gaussians (N-DG) (Diolatzis et al., 2024) extended to higher dimensions, refined by 6D Gaussian Splatting (6DGS) (Gao et al., 2025a) using conditional slicing. While elegant, Gaussian distributions cannot produce sharp angular cutoffs without many small primitives, limiting efficiency for specular materials.

**Alternative Kernel Designs.** Recent work explores kernels beyond standard 3D Gaussians. Several methods modify the kernel shape to better capture geometric structure: GES (Hamdi et al., 2024), TNT-GS (Liu et al., 2025b), Quadratic Gaussian Splatting (Zhang et al., 2025), 3D Half-Gaussian Splatting (Li et al., 2025), and Disc-GS (Qu et al., 2024) propose generalized exponentials, truncated, second-order, half, and discontinuity-aware Gaussian splats. Other approaches expand the distribution family, including Deformable Beta Splatting (DBS) (Liu et al., 2025a), 3D Student Splatting (Zhu et al., 2025), and Gabor Splats (Zhou et al., 2025) for sharper edges or heavier-tailed behavior. For appearance modeling, Textured Gaussians (Chao et al., 2025) enrich per-splat color detail. Beyond volumetric kernels, 3D Convex Splatting (Held et al., 2025b), Triangle Splatting (Held et al., 2025a), and Deformable Radial Kernel Splatting (Huang et al., 2025) introduce other geometric primitives as alternatives to ellipsoids. While these methods improve specific aspects of spatial reconstruction, they remain confined to 3D spatial kernels, without extending to unified high-dimensional splatting that jointly models spatial, angular, or temporal dimensions.

**Dynamic Neural Radiance Fields and Gaussian Splatting.** Early NeRF extensions modeled dynamics through deformation fields: D-NeRF (Pumarola et al., 2021) learns canonical-to-deformed mappings, Nerfies (Park et al., 2021a) handles non-rigid deformations, HyperNeRF (Park et al., 2021b) models topological changes, and HexPlane (Cao & Johnson, 2023) uses factorized representations.

For Gaussian Splatting, 4D Gaussian Splatting (4DGS) (Yang et al., 2024a) extends primitives temporally for real-time dynamic rendering. Dynamic 3D Gaussians (Luiten et al., 2024) and related works (Wu et al., 2024; Yang et al., 2024b) combine canonical representations with learned deformations. 7D Gaussian Splatting (7DGS) (Gao et al., 2025b) unifies spatial, temporal, and angular dimensions without separate deformation networks. Despite this unification, Gaussian's symmetric profile enforces smooth transitions—a single primitive cannot simultaneously represent sharp spatial edges, abrupt temporal motion, and narrow specular highlights.

Our Universal Beta Splatting addresses these limitations through controllable Beta kernels with per-dimension parameters, enabling dimension-specific adaptation and computational efficiency.

## 3 PRELIMINARY

**3D Gaussian Splatting (3DGS).** 3DGS (Kerbl et al., 2023) represents scenes as collections of anisotropic 3D Gaussians parameterized by position $\boldsymbol{\mu} \in \mathbb{R}^3$, opacity $o$, rotation quaternion $\boldsymbol{q}$, scale $\boldsymbol{s}$, and color features $\boldsymbol{f}$. The covariance matrix $\boldsymbol{\Sigma} = \boldsymbol{R}\boldsymbol{S}\boldsymbol{S}^\top\boldsymbol{R}^\top$ is constructed from rotation and scale matrices. To render an image, each 3D Gaussian ellipsoid is projected into the 2D image plane using the viewing transformation $\boldsymbol{W}$. This yields a 2D mean $\boldsymbol{\mu}' \in \mathbb{R}^2$ and a 2D covariance matrix $\boldsymbol{\Sigma}' = \boldsymbol{J}\boldsymbol{W}\boldsymbol{\Sigma}\boldsymbol{W}^\top\boldsymbol{J}^\top$, where $\boldsymbol{J}$ is the Jacobian of the projection. For a pixel location $\boldsymbol{x} \in \mathbb{R}^2$, the Mahalanobis distance to the projected splat is

$$r_i(\boldsymbol{x}) = (\boldsymbol{x} - \boldsymbol{\mu}_i')^\top \boldsymbol{\Sigma}_i'^{-1} (\boldsymbol{x} - \boldsymbol{\mu}_i'). \tag{1}$$

The Gaussian contribution to the pixel is then computed using a Gaussian kernel multiplied by opacity:

$$\sigma(\boldsymbol{x}) = \mathcal{G}(\boldsymbol{x}; \boldsymbol{\mu}, \boldsymbol{\Sigma}) \cdot o = e^{-\frac{1}{2}r_i(\boldsymbol{x})} \cdot o. \tag{2}$$

Finally, after sorting the primitives in front-to-back order, the pixel color is obtained using standard alpha compositing:

$$\boldsymbol{C}(\boldsymbol{x}) = \sum_{i=1}^{N} c_i \sigma_i(\boldsymbol{x}) \prod_{j=1}^{i-1} (1 - \sigma_j(\boldsymbol{x})). \tag{3}$$

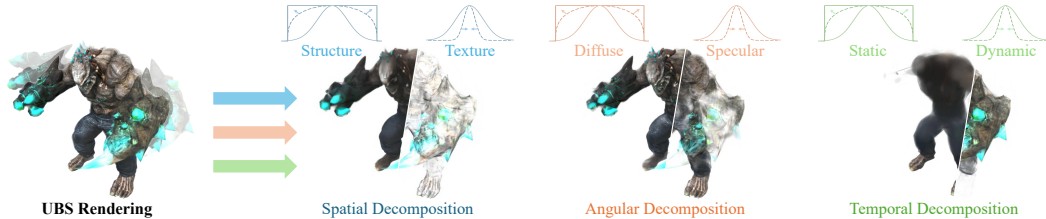

Figure 2: Visualization of decomposition. Our learned Beta parameters provide interpretable scene decomposition across spatial, angular, and temporal dimensions without explicit supervision.

**Deformable Beta Splatting (DBS).** DBS (Liu et al., 2025a) replaces Gaussian kernels with adaptive and bounded Beta kernels, defining density as $\sigma(\boldsymbol{x}) = \mathcal{B}(\boldsymbol{x}; \boldsymbol{\mu}, \boldsymbol{\Sigma}, b) \cdot o$, where

$$\mathcal{B}(\boldsymbol{x}; \boldsymbol{\mu}, \boldsymbol{\Sigma}, b) = (1 - r(\boldsymbol{x}))^{\beta(b)}, \quad \beta(b) = 4 \exp(b), \quad b \in \mathbb{R}, \quad r(\boldsymbol{x}) \in [0, 1). \tag{4}$$

The Beta kernel closely approximates a scaled Gaussian when initialization. As training progresses, the learnable parameter $b$ allows the kernel shape to adapt: negative $b$ produces flatter profiles with sharper boundaries suitable for solid geometry, while positive $b$ yields sharper peaks that better capture high-frequency details. This adaptability enables DBS to model diverse geometric structures more effectively than fixed Gaussian kernels. Appendix D includes more background and motivation.

**N-Dimensional Gaussian.** 6DGS (Gao et al., 2025a) and 7DGS (Gao et al., 2025b) extend 3DGS through conditional distributions. The key idea is to model a joint distribution over spatial and non-spatial dimensions (view/time), then condition on the query to obtain a view-specific or time-specific 3D Gaussian for rendering. 6DGS incorporates viewing direction into a 6D joint distribution:

$$X = \begin{pmatrix} X_p \\ X_d \end{pmatrix} \sim \mathcal{N}\left( \begin{pmatrix} \boldsymbol{\mu}_p \\ \boldsymbol{\mu}_d \end{pmatrix}, \begin{pmatrix} \boldsymbol{\Sigma}_p & \boldsymbol{\Sigma}_{pd} \\ \boldsymbol{\Sigma}_{pd}^\top & \boldsymbol{\Sigma}_d \end{pmatrix} \right). \tag{5}$$

Conditioning on view $d$ yields conditional covariance $\boldsymbol{\Sigma}_{\text{cond}} = \boldsymbol{\Sigma}_p - \boldsymbol{\Sigma}_{pd}\boldsymbol{\Sigma}_d^{-1}\boldsymbol{\Sigma}_{pd}^\top$, conditional mean $\boldsymbol{\mu}_{\text{cond}} = \boldsymbol{\mu}_p + \boldsymbol{\Sigma}_{pd}\boldsymbol{\Sigma}_d^{-1}(d - \boldsymbol{\mu}_d)$, and modulated opacity $o_{\text{cond}} = o \cdot \exp(-\lambda(d - \boldsymbol{\mu}_d)^\top \boldsymbol{\Sigma}_d^{-1}(d - \boldsymbol{\mu}_d))$.

7DGS adds temporal dimension $X_t$ for dynamic scenes, with conditioning on both time and view producing spatio-temporal-angular modulation. These methods establish high-dimensional representations but remain limited by fixed Gaussian profiles that cannot independently control cross-dimensional dependencies—a limitation UBS addresses through adaptive Beta kernels.

## 4 METHOD

In this section, we introduce Universal Beta Splatting (UBS), a unified framework that extends explicit radiance field rendering to $N$-dimensional anisotropic Beta kernels.

### 4.1 N-DIMENSIONAL BETA KERNEL

The core innovation of UBS is the $N$-dimensional anisotropic Beta kernel that generalizes the DBS density function to higher dimensions:

$$\sigma(\boldsymbol{x}, \boldsymbol{q}) = \mathcal{B}(\boldsymbol{x}, \boldsymbol{q}; \boldsymbol{\mu}, \boldsymbol{\Sigma}, \boldsymbol{b}) \cdot o, \tag{6}$$

where $\boldsymbol{x} \in \mathbb{R}^3$ represents spatial coordinates, the query $\boldsymbol{q} \in \mathbb{R}^{N-3}$ encodes additional dimensions (view direction, time, etc.), $\boldsymbol{\mu} \in \mathbb{R}^N$ is the mean vector, $\boldsymbol{\Sigma} \in \mathbb{R}^{N \times N}$ is the covariance matrix, $\boldsymbol{b} \in \mathbb{R}^{N-2}$ controls the Beta shape parameters across dimensions (Betas are shared in spatial domain), and $o \in [0, 1]$ is the opacity. The shape parameters $\boldsymbol{b}$ are transformed to ensure positive Beta exponents: $\beta_i = 4 \exp(b_i)$, enabling each dimension to adopt optimal kernel forms—from flat (negative $b_i$) to peaked (positive $b_i$).

In addition to the primitive shape, each kernel primitive also carries an RGB color $\boldsymbol{c} \in [0, 1]^3$. Crucially, because the Beta kernel can natively incorporate view and temporal dimensions through $\boldsymbol{q}$, no auxiliary color encoding is required to model view-dependent appearance or dynamics. This unified representation significantly reduces the per-primitive parameter count: for static scenes, UBS

achieves a 41% parameter reduction compared to 3DGS, and for dynamic scenes, UBS uses 73% fewer parameters than 4DGS, as detailed in Table 7 in the Appendix B. In contrast, existing methods rely on auxiliary encodings: 3DGS, 6DGS, and 7DGS employ 48-parameter spherical harmonics, 4DGS requires 144-parameter 4D spherical harmonics, and DBS needs additional spherical Beta functions for color modeling.

To maintain valid spatial geometry and guarantee backward compatibility while preserving high-dimensional expressiveness, we partition the $N$-dimensional space as:

$$\boldsymbol{\mu} = \begin{pmatrix} \boldsymbol{\mu}_x \\ \boldsymbol{\mu}_q \end{pmatrix}, \quad \boldsymbol{\Sigma} = \begin{pmatrix} \boldsymbol{\Sigma}_x & \boldsymbol{\Sigma}_{xq} \\ \boldsymbol{\Sigma}_{qx} & \boldsymbol{\Sigma}_q \end{pmatrix}, \tag{7}$$

where subscript $x$ denotes spatial components and $q$ denotes the additional dimensions. This partitioning allows conditioning on non-spatial dimensions to obtain renderable 3D representations.

## 4.2 SPATIAL-ORTHOGONAL CHOLESKY PARAMETERIZATION

While mean vectors, beta shapes, and colors are straightforward to parameterize, the covariance matrix requires additional care as it must be symmetric and positive semi-definite. In 3DGS, this is achieved by factorizing the covariance as $\boldsymbol{\Sigma} = \boldsymbol{R}\,\mathrm{diag}(\boldsymbol{s})^2\,\boldsymbol{R}^\top$, where $\boldsymbol{R}$ is a rotation matrix parameterized by quaternions and $\boldsymbol{s}$ are per-axis scales. However, quaternions do not generalize to higher dimensions ($N > 3$). $N$-dimensional Gaussian methods therefore resort to Cholesky factorization $\boldsymbol{\Sigma} = \boldsymbol{L}\boldsymbol{L}^\top$, which loses explicit orthogonality and often leads to suboptimal solutions in the spatial subspace.

To address this limitation, we introduce the *Spatial-Orthogonal Cholesky* parameterization that preserves rotation-scale structure in the spatial subspace while enabling flexible cross-dimensional correlations. For $N = 3 + C$ dimensions, we construct:

$$\boldsymbol{L} = \begin{pmatrix} \boldsymbol{R}_x\,\mathrm{diag}(\boldsymbol{s}_x) & \boldsymbol{0} \\ \boldsymbol{L}_{qx} & \boldsymbol{L}_q \end{pmatrix}, \tag{8}$$

where $\boldsymbol{R}_x \in \mathrm{SO}(3)$ is the spatial rotation computed via first-order Taylor approximation $\boldsymbol{R}_x \approx \boldsymbol{I} + \boldsymbol{A}_x$ of the matrix exponential (with $\boldsymbol{A}_x$ skew-symmetric), $\boldsymbol{s}_x$ are spatial scales, $\boldsymbol{s}_q$ are scales for other dimensions, and $\boldsymbol{L}_{qx}$ encodes cross-correlations. As shown in Table 10 (Appendix C), this first-order approximation provides nearly identical fidelity while greatly reducing computation. The covariance $\boldsymbol{\Sigma} = \boldsymbol{L}\boldsymbol{L}^\top$ maintains explicit 3D geometric structure while allowing expressive correlations across view and time. More theoretical and empirical analysis are available in Appendix C.

## 4.3 BETA-MODULATED CONDITIONAL SLICING

In $N$-dimensional Gaussians, all dimensions are coupled: when the input view or time changes, the primitive is inevitably dragged along—it shifts in space, stretches in shape, and fades in opacity. To break this constraint, we introduce Beta-Modulated Conditional Slicing.

To render at query condition $\boldsymbol{q}$, we condition the $N$-dimensional Beta kernel to obtain a 3D spatial representation. We introduce Beta modulation to the standard conditional Gaussian formulation:

**Beta Parameters.** We define per-dimension Beta exponents with spatial parameter sharing: $\boldsymbol{\beta} = [\beta_x, \beta_x, \beta_x, \beta_{q_1}, \ldots, \beta_{q_C}]$ where $\beta_x = 4\exp(b_x)$ is shared across the first three spatial dimensions and $\beta_{q_i} = \exp(b_{q_i})$ are individual parameters for the $C$ non-spatial dimensions. We share the spatial Beta parameter across all three spatial dimensions to ensure that splatted kernels maintain consistent spatial geometry. This design choice prevents inconsistent spatial artifacts while still allowing per-dimension control in angular and temporal domains.

**Beta-Modulated Conditioning.** The conditional mean and covariance incorporate dimension-specific Beta modulation:

$$\boldsymbol{\mu}_{x|q} = \boldsymbol{\mu}_x + \boldsymbol{\Sigma}_{xq}\,\boldsymbol{\Sigma}_q^{-1}\,\mathrm{diag}(\tilde{\boldsymbol{\beta}}_q)\,(\boldsymbol{q} - \boldsymbol{\mu}_q), \tag{9}$$

$$\boldsymbol{\Sigma}_{x|q} = \boldsymbol{\Sigma}_x - \boldsymbol{\Sigma}_{xq}\,\boldsymbol{\Sigma}_q^{-1}\,\mathrm{diag}(\tilde{\boldsymbol{\beta}}_q)\,\boldsymbol{\Sigma}_{qx}, \tag{10}$$

where $\mathrm{diag}(\tilde{\boldsymbol{\beta}}_q) = \mathrm{diag}([\beta_{q_1}, \ldots, \beta_{q_C}]_{\leq 1})$ applies dimension-specific Beta modulation to the non-spatial conditioning terms while preserving isotropic spatial structure.

**Beta-Modulated Opacity.** We adopt a factorized opacity with inherent multidimensional support:

$$d_i^{\text{raw}} = (\hat{\boldsymbol{L}}_q^{-1}(\boldsymbol{q} - \boldsymbol{\mu}_q))_i^2, \quad d_i = \tanh(d_i^{\text{raw}}) \in [0,1), \quad o(\boldsymbol{q}) = o \prod_{i=1}^{C}(1 - d_i)^{4\beta_{q_i}}, \quad (11)$$

where $\hat{\boldsymbol{L}}_q$ denotes the Cholesky factor of $\boldsymbol{\Sigma}_q$, such that $\hat{\boldsymbol{L}}_q^{-1}\hat{\boldsymbol{L}}_q^{-\top} = \boldsymbol{\Sigma}_q^{-1}$. Then $\tanh$ maps per-dimension Mahalanobis distance $d_i^{\text{raw}}$ into a bounded value $d_i \in [0,1)$, satisfying the Beta kernel input definition. The modulated product form allows independent control over each non-spatial dimension's contribution.

**Final Beta-Splat Density.** The conditioned 3D Beta kernel becomes:

$$\sigma(\boldsymbol{x}, \boldsymbol{q}) = \mathcal{B}\big(\boldsymbol{x}; \boldsymbol{\mu}_{x|q}, \boldsymbol{\Sigma}_{x|q}, b_x\big) \cdot o(\boldsymbol{q}), \quad (12)$$

where the spatial Beta $b_x$ controls the 3D kernel shape uniformly across all spatial dimensions.

### 4.4 UNIVERSAL COMPATIBILITY AND INTERPRETABILITY

**Adaptive Specialization.** UBS naturally specializes for different applications through its query dimensions: (1) $\boldsymbol{q} = \boldsymbol{d}$ (view direction, $N = 6$) for view-dependent rendering with anisotropic BRDF modeling, denoted as UBS-6D; (2) $\boldsymbol{q} = [t, \boldsymbol{d}]$ (time + view, $N = 7$) for dynamic scenes capturing spatio-temporal-angular correlations, denoted as UBS-7D; and (3) seamless extension to higher dimensions for additional modalities like lighting or material properties.

**Universal Compatibility.** UBS provides backward compatibility by design, reducing to existing methods under specific parameter settings. For $N = 3$ (pure spatial), UBS reduces to Deformable Beta Splatting with $\sigma(\boldsymbol{x}) = \mathcal{B}(\boldsymbol{x}; \boldsymbol{\mu}_x, \boldsymbol{\Sigma}_x, b_x) \cdot o$, and further approximates to standard 3DGS when $b_x = 0$ (Gaussian limit). For higher dimensions with all Beta parameters zero ($\boldsymbol{b} = \boldsymbol{0}$), UBS approximates corresponding Gaussian methods: 6DGS for view-dependent rendering ($N = 6$) and 7DGS for dynamic scenes ($N = 7$). This guaranteed compatibility ensures UBS serves as a seamless drop-in replacement while providing performance lower bounds.

**Interpretable Decomposition.** Beyond compatibility, learned Beta parameters provide interpretable scene decomposition without explicit supervision. Spatial parameters distinguish geometry types: negative $b_x$ produces flat kernels for smooth surfaces, while positive $b_x$ creates peaked kernels for fine textures. Angular parameters separate appearance models: negative $b_d$ yields broad responses for diffuse surfaces, while positive $b_d$ generates sharp peaks for specular reflections. Temporal parameters isolate motion patterns: negative $b_t$ maintains broad support for static elements, while positive $b_t$ provides localized activation for dynamic components, as illustrated in Figure 2. This decomposition enables applications like relighting and motion analysis without requiring separate encodings or auxiliary networks.

### 4.5 OPTIMIZATION AND IMPLEMENTATION

For primitive management, we adopt the kernel-agnostic MCMC optimization from DBS (Liu et al., 2025a), which provides stable, distribution-preserving updates independent of kernel form.

**Loss Function.** We optimize with reconstruction losses and regularization:

$$\mathcal{L} = (1 - \lambda_{\text{SSIM}})\mathcal{L}_1 + \lambda_{\text{SSIM}}\mathcal{L}_{\text{SSIM}} + \lambda_o \sum_i |o_i| + \lambda_\Sigma \sum_i ||\boldsymbol{s}_i||_1, \quad (13)$$

where opacity regularization $\lambda_o$ ensures valid MCMC densification and scale penalty $\lambda_\Sigma$ encourages primitive relocation. Spatial noise injection $\boldsymbol{\mu}_x \leftarrow \boldsymbol{\mu}_x - \lambda_{\text{lr}}\nabla_{\boldsymbol{\mu}}\mathcal{L} + \lambda_\epsilon \epsilon$ promotes exploration.

**Densification.** Cloned primitive opacity follows $o' = 1 - \sqrt[N]{1 - o}$ to preserve distribution. Under opacity regularization, adjusting opacity maintains density distribution regardless of kernel type.

**CUDA Implementation.** We provide fused CUDA kernels for Beta evaluation, conditional slicing, and spatial-orthogonal operations. The pipeline maintains compatibility with existing frameworks: $N$-dimensional kernels are conditioned to 3D then rasterized using standard splatting. This modular design enables real-time performance while allowing future integration with acceleration techniques like FlashGS (Feng et al., 2025) and extended camera models from 3DGUT (Wu et al., 2025).

## 5 EXPERIMENTS

### 5.1 EXPERIMENTAL SETUP

**Datasets.** We evaluate UBS across five diverse datasets covering both static and dynamic scenes:

- *Static:* we use (1) *NeRF Synthetic* (Mildenhall et al., 2020) with 8 synthetic scenes under controlled lighting for geometric reconstruction assessment; (2) *Mip-NeRF 360* (Barron et al., 2022) sharing 9 unbounded real-world scenes challenging anti-aliasing and view synthesis; and (3) *6DGS-PBR* (Gao et al., 2025a) featuring 7 physically-based scenes with complex view-dependent effects including volumetric scattering, translucent materials, subsurface scattering, and medical volumetrics.
- *Dynamic:* we evaluate on: (4) *D-NeRF* (Pumarola et al., 2021) with 8 synthetic sequences featuring various deformation types; and (5) *7DGS-PBR* (Gao et al., 2025b) containing 6 physically-based dynamic scenes with complex spatio-temporal-angular correlations including cardiac motion, daylight transitions, animated volumetric effects, and translucent deformations.

**Evaluation Metrics.** We assess reconstruction quality using standard metrics: Peak Signal-to-Noise Ratio (PSNR), Structural Similarity Index Measure (SSIM) (Wang et al., 2004), and Learned Perceptual Image Patch Similarity (LPIPS) (Zhang et al., 2018). Additionally, we report rendering speed (FPS) and training time to evaluate computational efficiency. All metrics are computed on held-out test views following standard evaluation protocols.

**Implementation Details.** UBS is implemented in PyTorch with custom CUDA kernels for Beta kernel evaluation and conditional slicing operations. We conduct on a single NVIDIA RTX 4090 24GB GPU for static and ablation experiments and a single NVIDIA Tesla V100 16GB GPU for dynamic experiments to be consistent 7DGS. Training employs the Adam optimizer for 30K iterations with learning rates of $1.6 \times 10^{-4}$ for positions, $5 \times 10^{-2}$ for opacity, $5 \times 10^{-3}$ for scales, and $1 \times 10^{-3}$ for other parameters. The regularization weights are set to $\lambda_o = 0.01$, $\lambda_\Sigma = 0.01$, and $\lambda_\epsilon = 1$. For $N$-dimensional cases, we randomly initialize temporal and directional means ($\mu_t$, $\boldsymbol{\mu}_d$) within [0, 1]. Beta parameters are initialized to zero, corresponding to the Gaussian limit for guaranteed convergence. For dynamic scenes, we use a batch size of 4 to maintain consistency with 4DGS and 7DGS baselines. Unlike these methods which evaluate on test sets every 500 iterations and report best results in their papers, we report at the final iteration (30K) to demonstrate convergence stability.

### 5.2 COMPARISON WITH THE STATE OF THE ART

Table 1 compares our UBS with 3DGS (Kerbl et al., 2023), 4DGS (Yang et al., 2024a), 6DGS (Gao et al., 2025a), and 7DGS (Gao et al., 2025b) on both static and dynamic benchmarks.

**Static Scenes.** Our method UBS-6D consistently achieves superior reconstruction quality across all datasets while maintaining competitive training efficiency on three static scene benchmarks:

- *NeRF Synthetic:* UBS-6D achieves 34.92 dB PSNR, improving by +1.13 dB over 3DGS and +1.32 dB over 6DGS. Despite lacking strong view-dependent effects, the dataset benefits from Beta kernels' geometric adaptivity, with notable improvements on scenes requiring diverse surface representations (`ficus`: +1.41 dB over 3DGS, `materials`: +2.40 dB over 3DGS).
- *Mip-NeRF 360:* UBS-6D reaches 28.66 dB PSNR, surpassing 3DGS by +1.23 dB and 6DGS by +2.31 dB. The real-world unbounded scenes showcase UBS's effectiveness on complex lighting conditions, with substantial gains on indoor environments (`counter`: +2.43 dB over 3DGS, `bonsai`: +3.00 dB over 3DGS).
- *6DGS-PBR:* UBS-6D demonstrates the most significant improvements with 40.10 dB PSNR (+12.70 dB over 3DGS, +3.42 dB over 6DGS average). The physically-based scenes with volumetric scattering and translucency particularly benefit from per-dimension Beta control: volumetric effects like `cloud` improve by +5.81 dB over 6DGS, while complex materials like `bunny` achieve a remarkable +8.27 dB improvement over 6DGS.

Overall, UBS-6D demonstrates exceptional performance on challenging view-dependent scenarios in *6DGS-PBR*. In addition, UBS-6D establishes new state-of-the-art results on *NeRF Synthetic* and *Mip-NeRF 360* shown in Table 4 in Appendix A.

Table 1: Comparison with 3DGS and 6DGS on static benchmarks of NeRF Synthetic (Mildenhall et al., 2020), Mip-NeRF (Barron et al., 2022), 6DGS-PBR (Gao et al., 2025a), and with 4DGS and 7DGS on dynamic benchmarks of 7DGS-PBR (Gao et al., 2025b) and D-NeRF (Pumarola et al., 2021). 'Train' means time in mins.

| Dataset | | Scene | 3DGS | | | | 6DGS | | | | UBS-6D (Ours) | | | |
|---|---|---|---|---|---|---|---|---|---|---|---|---|---|---|
| | | | PSNR↑ | SSIM↑ | LPIPS↓ | Train↓ | PSNR↑ | SSIM↑ | LPIPS↓ | Train↓ | PSNR↑ | SSIM↑ | LPIPS↓ | Train↓ |
| STATIC SCENES | NeRF Synthetic | chair | 35.60 | 0.988 | 0.010 | **4.8** | 35.55 | 0.986 | 0.011 | 19.6 | **36.72** | **0.990** | **0.009** | 8.7 |
| | | drums | 26.28 | 0.955 | 0.037 | **4.1** | 26.63 | 0.951 | 0.038 | 15.4 | **27.19** | **0.960** | **0.031** | 8.1 |
| | | ficus | 35.49 | 0.987 | 0.012 | **3.0** | 34.62 | 0.984 | 0.015 | 11.3 | **36.90** | **0.990** | **0.009** | 8.2 |
| | | hotdog | 38.07 | 0.985 | 0.020 | **3.5** | 37.96 | 0.983 | 0.021 | 10.0 | **38.67** | **0.988** | **0.016** | 9.1 |
| | | lego | 36.06 | 0.983 | 0.016 | **4.0** | 35.22 | 0.979 | 0.020 | 15.3 | **36.95** | **0.985** | **0.014** | 8.4 |
| | | materials | 30.50 | 0.960 | 0.037 | **2.7** | 30.63 | 0.960 | 0.041 | 7.8 | **32.90** | **0.977** | **0.025** | 8.9 |
| | | mic | 36.67 | 0.992 | 0.006 | **3.0** | 37.10 | 0.992 | 0.007 | 9.4 | **37.86** | **0.994** | **0.005** | 9.5 |
| | | ship | 31.68 | 0.906 | 0.106 | **4.1** | 31.09 | 0.899 | 0.118 | 12.7 | **32.13** | **0.914** | **0.100** | 8.8 |
| | | avg | 33.79 | 0.970 | 0.030 | **3.6** | 33.60 | 0.967 | 0.034 | 12.7 | **34.92** | **0.975** | **0.026** | 8.7 |
| | Mip-NeRF 360 | bicycle | 25.19 | 0.764 | 0.212 | **29.2** | 22.17 | 0.535 | 0.425 | 114.9 | **25.64** | **0.793** | **0.186** | 30.2 |
| | | flowers | 21.39 | 0.604 | 0.338 | **19.6** | 18.99 | 0.417 | 0.461 | 88.8 | **21.96** | **0.635** | **0.313** | **19.6** |
| | | garden | 27.27 | 0.862 | 0.109 | 30.4 | 26.71 | 0.843 | 0.142 | 164.9 | **28.10** | **0.883** | **0.094** | 36.8 |
| | | stump | 26.61 | 0.772 | 0.215 | 22.7 | 23.83 | 0.665 | 0.338 | 60.2 | **26.75** | **0.788** | **0.199** | **18.9** |
| | | treehill | 22.47 | 0.631 | 0.329 | **20.1** | 21.51 | 0.556 | 0.425 | 94.0 | **23.57** | **0.684** | **0.280** | 21.9 |
| | | room | 31.49 | 0.917 | 0.221 | 18.2 | 31.34 | 0.904 | 0.256 | 51.4 | **32.81** | **0.940** | **0.173** | 21.9 |
| | | counter | 28.96 | 0.906 | 0.202 | **17.6** | 29.17 | 0.880 | 0.253 | 43.5 | **31.39** | **0.937** | **0.150** | 27.8 |
| | | kitchen | 31.40 | 0.926 | 0.127 | **21.5** | 30.48 | 0.907 | 0.158 | 71.3 | **32.64** | **0.940** | **0.107** | 26.6 |
| | | bonsai | 32.05 | 0.940 | 0.206 | **15.7** | 32.91 | 0.935 | 0.228 | 48.4 | **35.05** | **0.962** | **0.154** | 22.7 |
| | | avg | 27.43 | 0.814 | 0.218 | **21.7** | 26.35 | 0.738 | 0.298 | 81.9 | **28.66** | **0.840** | **0.184** | 25.2 |
| | 6DGS-PBR | bunny | 29.19 | 0.985 | 0.074 | **22.8** | 37.41 | 0.991 | 0.052 | 24.1 | **45.68** | **0.996** | **0.023** | 32.1 |
| | | cloud | 34.03 | 0.985 | 0.058 | 23.1 | 41.00 | 0.991 | 0.050 | **21.6** | **46.81** | **0.995** | **0.027** | 30.6 |
| | | explosion | 27.06 | 0.953 | 0.097 | **11.3** | 41.61 | 0.990 | 0.031 | 18.3 | **44.54** | **0.994** | **0.018** | 32.4 |
| | | smoke | 26.82 | 0.964 | 0.088 | **24.2** | 41.41 | 0.993 | 0.041 | 31.1 | **44.02** | **0.995** | **0.028** | 33.5 |
| | | suzanne | 22.45 | 0.885 | 0.159 | **19.5** | 26.96 | 0.928 | 0.106 | 32.9 | **27.35** | **0.931** | **0.098** | 35.1 |
| | | dragon | 26.53 | 0.812 | 0.123 | **12.0** | 34.95 | 0.933 | 0.044 | 16.3 | **38.06** | **0.974** | **0.034** | 23.7 |
| | | ct-scan | 25.69 | 0.917 | 0.099 | **4.5** | 33.46 | 0.964 | 0.058 | 11.0 | **34.24** | **0.969** | **0.046** | 6.5 |
| | | avg | 27.40 | 0.929 | 0.100 | **16.8** | 36.68 | 0.970 | 0.055 | 22.2 | **40.10** | **0.979** | **0.039** | 27.7 |

| Dataset | | Scene | 4DGS | | | | 7DGS | | | | UBS-7D (Ours) | | | |
|---|---|---|---|---|---|---|---|---|---|---|---|---|---|---|
| | | | PSNR↑ | SSIM↑ | LPIPS↓ | Train↓ | PSNR↑ | SSIM↑ | LPIPS↓ | Train↓ | PSNR↑ | SSIM↑ | LPIPS↓ | Train↓ |
| DYNAMIC SCENES | 7DGS-PBR | heart1 | 27.23 | 0.949 | 0.046 | 103.0 | 34.66 | 0.983 | 0.023 | 114.2 | **37.44** | **0.990** | **0.013** | 38.7 |
| | | heart2 | 25.09 | 0.919 | 0.085 | 103.4 | 30.99 | 0.959 | 0.057 | 384.6 | **31.99** | **0.966** | **0.043** | 33.5 |
| | | cloud | 24.63 | 0.938 | 0.100 | 123.7 | 29.28 | 0.955 | 0.075 | 102.6 | **30.65** | **0.955** | **0.066** | 50.0 |
| | | dust | 35.81 | 0.954 | 0.037 | 97.0 | 36.85 | 0.955 | 0.038 | 69.8 | **39.17** | **0.985** | **0.028** | 46.5 |
| | | flame | 29.25 | 0.927 | 0.068 | 113.7 | 31.64 | 0.937 | 0.062 | 74.1 | **31.70** | **0.963** | **0.057** | 53.7 |
| | | suzanne | 24.41 | 0.912 | 0.127 | 222.5 | 27.09 | **0.941** | 0.072 | 193.9 | **27.12** | 0.934 | **0.069** | 136.8 |
| | | avg | 27.74 | 0.933 | 0.077 | 127.2 | 31.75 | 0.955 | 0.055 | 116.7 | **33.00** | **0.966** | **0.046** | 59.9 |
| | D-NeRF | b.balls | 32.45 | 0.979 | 0.028 | 50.7 | **33.61** | 0.980 | 0.025 | 86.6 | 33.39 | **0.982** | **0.022** | 49.6 |
| | | h.warrior | 33.69 | 0.944 | 0.069 | 35.3 | 32.30 | 0.931 | 0.088 | **31.3** | **33.81** | **0.934** | **0.086** | 36.3 |
| | | hook | 31.90 | 0.965 | 0.037 | 38.0 | 30.19 | 0.953 | 0.047 | 35.7 | **30.93** | **0.956** | **0.045** | **34.0** |
| | | j.jacks | 28.43 | 0.962 | 0.041 | 66.9 | **31.37** | **0.967** | **0.038** | 34.1 | 30.96 | 0.966 | 0.040 | 43.2 |
| | | lego | 24.28 | 0.903 | 0.098 | 55.4 | 27.64 | **0.935** | **0.067** | 78.5 | **27.85** | 0.925 | 0.072 | **31.3** |
| | | mutant | 38.64 | 0.990 | 0.009 | **39.1** | 39.21 | 0.992 | 0.008 | 42.5 | **40.75** | **0.994** | **0.006** | 60.0 |
| | | standup | 39.00 | 0.990 | 0.009 | 34.4 | 38.42 | 0.987 | 0.014 | **33.5** | **39.08** | **0.991** | **0.009** | 41.1 |
| | | trex | 28.35 | 0.973 | 0.026 | 100.7 | 29.94 | **0.978** | **0.021** | 63.4 | **30.05** | 0.975 | 0.023 | **41.6** |
| | | avg | 32.09 | 0.963 | 0.040 | 52.6 | 32.84 | 0.965 | 0.039 | 50.7 | **33.35** | **0.965** | **0.038** | 42.1 |

**Dynamic Scenes.** We compare our method UBS-7D with 4DGS and 7DGS on 2 dynamic benchmarks. Note that we consider the base 7DGS without MLP networks for adaptive Gaussian refinement (AGR) to ensure fair comparison and isolate the benefits of our primitive design from auxiliary networks:

- *7DGS-PBR:* UBS-7D achieves 33.00 dB PSNR, outperforming 4DGS by +5.26 dB and 7DGS by +1.25 dB. The physically-based dynamic scenes with spatio-temporal-angular correlations show substantial improvements, particularly on complex volumetric dynamics (dust: +2.32 dB over 7DGS) and cardiac motion sequences (heart1: +2.78 dB over 7DGS).
- *D-NeRF:* UBS-7D reaches 33.35 dB PSNR (+1.26 dB over 4DGS, +0.51 dB over 7DGS). The synthetic monocular sequences benefit from Beta kernels' ability to model diverse temporal behaviors, with notable gains on scenes requiring both spatial detail and temporal precision (mutant: +1.54 dB over 7DGS, h.warrior: +1.51 dB over 7DGS).

The consistent improvements across PSNR, SSIM, and LPIPS metrics confirm that UBS's enhanced representational capacity translates to comprehensive quality gains. Figure 3 demonstrates that UBS

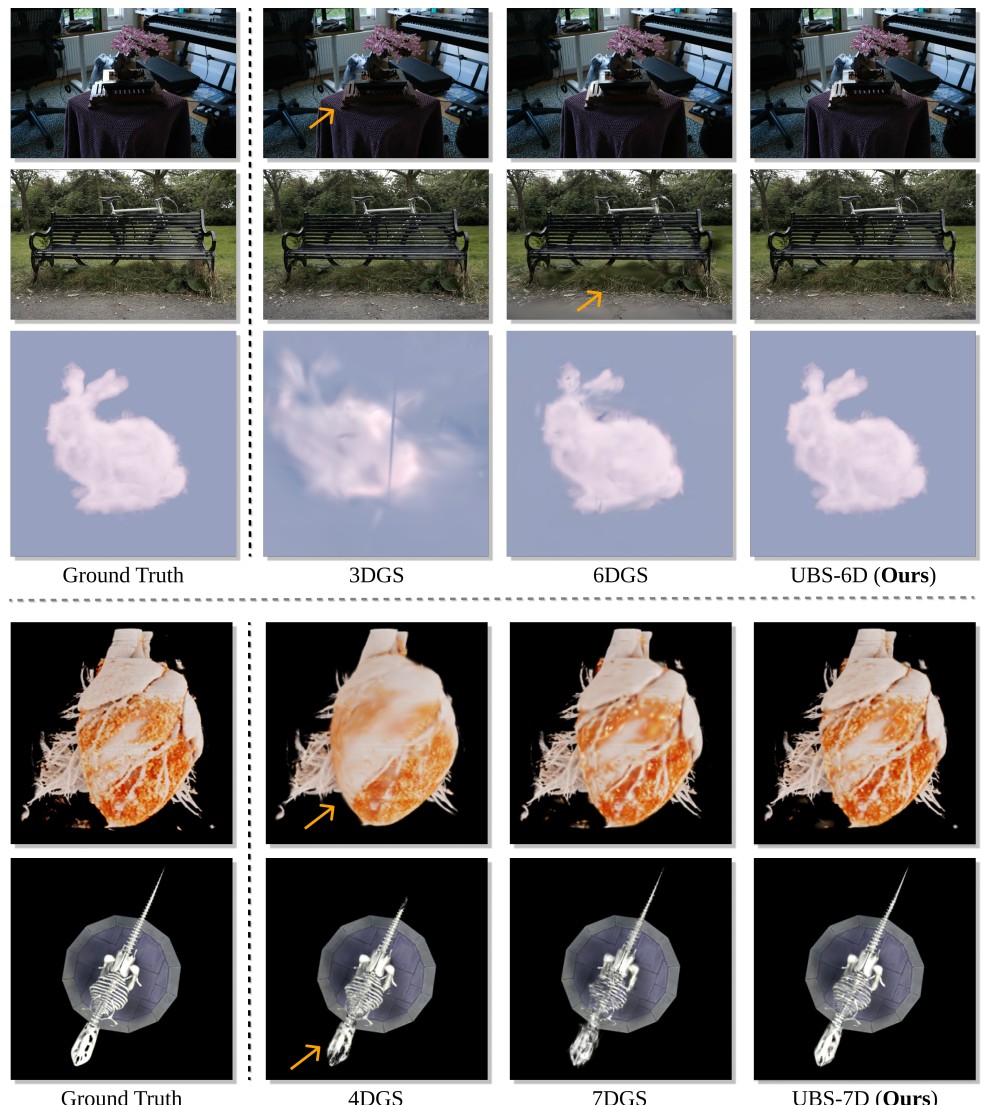

Figure 3: Qualitative comparison of methods for static and dynamic scenes.

achieves better visual quality in both static and dynamic scenes compared with baseline methods. Additional quantitative benchmark comparisons are provided in Appendix A.

**Efficiency Analysis.** Beyond reconstruction quality, UBS demonstrates superior computational efficiency through its streamlined primitive design and optimized CUDA implementation. Table 2 summarizes the efficiency comparison across all benchmarks, with detailed per-scene results provided in Table 8 in the Appendix B. For static scenes, UBS-6D achieves significantly faster training than 6DGS on NeRF Synthetic (8.7 vs 12.7 minutes, 31.5% reduction) and Mip-NeRF 360 (25.2 vs 81.9 minutes, 69.2% reduction) with competitive memory usage. On 6DGS-PBR, UBS-6D requires longer training time (27.7 vs 22.2 minutes) due to our use of higher primitive counts (300K vs 68K) to fully exploit Beta kernels' representational capacity for complex volumetric effects. For dynamic scenes, UBS-7D delivers substantial training time reductions of 48.7% on 7DGS-PBR (59.9 vs 116.7 minutes) and 17.0% on D-NeRF (42.1 vs 50.7 minutes), both compared to 7DGS. These efficiency gains stem from our unified Beta kernel approach, which eliminates auxiliary spherical harmonic encodings while enabling efficient CUDA-accelerated conditional slicing operations.

Table 2: Efficiency comparison on static and dynamic benchmarks (averaged across scenes). '*Train*' indicates training time in minutes; '*Mem*' indicates storage size in MB. Detailed per-scene results and per-primitive parameter comparisons are provided in Appendix B.

| Dataset | 3DGS/4DGS | | | | 6DGS/7DGS | | | | UBS (**Ours**) | | | |
|---|---|---|---|---|---|---|---|---|---|---|---|---|
| | FPS↑ | Num↓ | Mem↓ | Train↓ | FPS↑ | Num↓ | Mem↓ | Train↓ | FPS↑ | Num↓ | Mem↓ | Train↓ |
| NeRF Synthetic | 695.6 | 289.3K | 68.5 | 3.6 | 648.0 | 240.7K | 69.8 | 12.7 | 345.7 | 300.0K | 40.1 | 8.7 |
| Mip-NeRF 360 | 178.7 | 3.3M | 775.9 | 21.7 | 157.4 | 2.0M | 581.2 | 81.9 | 100.4 | 3.1M | 409.0 | 25.2 |
| 6DGS-PBR | 305.9 | 147.7K | 34.9 | 16.8 | 339.8 | 84.3K | 24.4 | 22.2 | 290.2 | 300.0K | 40.1 | 27.7 |
| 7DGS-PBR | 192.6 | 642.0K | 613.1 | 127.2 | 376.0 | 88.4K | 29.7 | 116.7 | 359.6 | 234.3K | 39.4 | 59.9 |
| D-NeRF | 296.4 | 255.3K | 380.3 | 52.6 | 377.8 | 43.8K | 16.0 | 50.7 | 368.2 | 168.8K | 28.4 | 42.1 |

Table 3: Ablation study of UBS across representative static and dynamic scenes.

| | Static Scenes | | | | | | Dynamic Scenes | | | | | | Mean | | |
|---|---|---|---|---|---|---|---|---|---|---|---|---|---|---|---|
| | ficus | | | bonsai | | | cloud | | | lego | | | | | |
| | PSNR↑ | Train↓ | FPS↑ | PSNR↑ | Train↓ | FPS↑ | PSNR↑ | Train↓ | FPS↑ | PSNR↑ | Train↓ | FPS↑ | PSNR↑ | Train↓ | FPS↑ |
| skew-sym. w/o Cholesky | 36.34 | 19.0 | 207.2 | 34.24 | 71.1 | 50.1 | 28.49 | 14.7 | 142.5 | 28.42 | 13.1 | 241.9 | 31.87 | 29.5 | 160.4 |
| Cholesky w/o spatial-ortho. | 35.76 | 22.8 | 119.2 | 34.21 | 75.0 | 40.4 | 28.41 | 16.3 | 162.2 | 28.02 | 15.0 | 175.4 | 31.60 | 32.3 | 124.3 |
| w/o Beta condition | 36.72 | 13.7 | 217.2 | 34.30 | 49.3 | 53.9 | 28.26 | 13.7 | 160.0 | 27.22 | 11.4 | 198.9 | 31.63 | 22.0 | 157.5 |
| w/o Beta opacity | 33.76 | 17.9 | 213.0 | 33.17 | 62.1 | 57.3 | 21.93 | 11.3 | **260.3** | 25.68 | 11.9 | 209.5 | 28.64 | 25.8 | 185.0 |
| full | 36.92 | 20.7 | 188.4 | 34.62 | 72.9 | 48.9 | **28.61** | 16.1 | 158.3 | 28.47 | 13.1 | 227.6 | 32.16 | 30.7 | 155.8 |
| full w/ CUDA | **36.96** | **9.3** | **262.4** | **34.65** | **24.1** | **78.5** | 28.58 | **10.2** | 184.8 | **28.55** | **7.8** | **259.1** | **32.19** | **12.9** | **196.2** |

## 5.3 ABLATION STUDY

We conduct ablations on representative samples of static scenes in *6DGS-PBR* (Gao et al., 2025a) dataset (ficus, bonsai) and of dynamic scenes in *7DGS-PBR* (Gao et al., 2025b) dataset (cloud, lego) to validate our design choices and efficient CUDA implementation (Table 3).

**Spatial-Orthogonal Parameterization.** Pure rotation-scale parameterization and full Cholesky without spatial orthogonality reduce quality by 0.29 dB and 0.56 dB, respectively, confirming the importance of preserving the spatial SO(3) structure and maintaining flexibility of other dimensions. **Beta-Modulated Conditioning.** Removing Beta modulation decreases performance by 0.53 dB, with larger drops on dynamic scenes (lego: $-1.25$ dB), validating dimension-specific shape control for decoupling spatial-angular-temporal responses. **Beta-Modulated Opacity.** Disabling Beta-modulated opacity causes the most significant degradation ($-3.52$ dB average), particularly on volumetric scenes (cloud: $-6.68$ dB), proving the gated product form opacity crucial for anisotropic falloff. **CUDA Acceleration.** Custom CUDA kernels reduce training time by 58% (12.9 vs 30.7 minutes) and improve rendering speed by 26% (196.2 vs 155.8 FPS), enhancing real-time performance through fused Beta evaluation and conditional slicing operations.

These studies confirm each component's contribution to UBS's performance, with the complete model showing consistent gains across diverse scenarios while maintaining computational efficiency.

## 6 CONCLUSION

We presented Universal Beta Splatting (UBS), a unified framework extending radiance field rendering to $N$-dimensional anisotropic Beta kernels. By replacing fixed Gaussian profiles with adaptive per-dimension shape control, UBS simultaneously models spatial geometry, view-dependent appearance, and temporal dynamics within a single primitive. Our spatial-orthogonal parameterization and Beta-modulated conditional slicing enable independent dimensional control while preserving geometric consistency. Experiments demonstrate significant improvements—up to +8.27 dB PSNR on volumetric scenes and 48.7% faster training on dynamic sequences—while maintaining backward compatibility by approximating Gaussian methods. The learned Beta parameters provide interpretable scene decomposition without explicit supervision, and our CUDA implementation achieves real-time rendering, establishing Beta kernels as practical universal primitives for radiance fields.

ACKNOWLEDGMENTS

This work was supported in part by the U.S. Army Combat Capabilities Development Command under contract numbers W911NF-14-D-0005 and W912CG-24-D-0001. The authors thank Mr. Clayton Burford of the Battlespace Content Creation (BCC) team at Simulation and Training Technology Center (STTC). The content of the information does not necessarily reflect the position or the policy of the Government, and no official endorsement should be inferred.

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

# Universal Beta Splatting

## Supplementary Material

## A  MORE COMPARISONS

### A.1  COMPARISON WITH STATE-OF-THE-ART METHODS

Table 4 presents a comprehensive comparison of UBS-6D with state-of-the-art neural rendering methods on the Mip-NeRF360 (Barron et al., 2022) and NeRF Synthetic (Mildenhall et al., 2020) benchmarks. Our method achieves the highest PSNR on both datasets, demonstrating the effectiveness of adaptive Beta kernels over fixed Gaussian profiles. The consistent improvements across both synthetic and real-world scenes validate the universal applicability of our Beta kernel framework.

Table 4: Comparisonh with SOTA methods on Mip-NeRF360 (Barron et al., 2022) and NeRF Synthetic (Mildenhall et al., 2020).

| | Methods | Mip-NeRF360 | | | NeRF Synthetic | | |
|---|---|---|---|---|---|---|---|
| | | PSNR↑ | SSIM↑ | LPIPS↓ | PSNR↑ | SSIM↑ | LPIPS↓ |
| implicit | Instant-NGP (Müller et al., 2022) | 25.51 | 0.684 | 0.398 | 33.18 | 0.959 | 0.055 |
| | Mip-NeRF360 (Barron et al., 2022) | 27.69 | 0.792 | 0.237 | 33.25 | 0.962 | 0.039 |
| | Zip-NeRF (Barron et al., 2023) | 28.54 | 0.828 | 0.189 | 33.10 | 0.971 | 0.031 |
| explicit | 3DGS (Kerbl et al., 2023) | 27.20 | 0.815 | 0.214 | 33.31 | 0.969 | 0.037 |
| | GES (Hamdi et al., 2024) | 26.91 | 0.794 | 0.250 | - | - | - |
| | 2DGS (Huang et al., 2024) | 27.04 | 0.805 | 0.297 | 33.07 | - | - |
| | Mip-Splatting (Yu et al., 2024) | 27.79 | 0.827 | 0.203 | 33.33 | 0.969 | 0.039 |
| | 3DGS-MCMC (Kheradmand et al., 2024) | 28.29 | 0.840 | 0.210 | 33.80 | 0.970 | 0.040 |
| | DRKS (Huang et al., 2025) | 26.76 | 0.787 | 0.236 | 33.82 | - | - |
| | Textured GS (Chao et al., 2025) | 27.35 | 0.827 | 0.186 | 33.24 | 0.967 | 0.043 |
| | Quadratic GS (Zhang et al., 2025) | 27.39 | 0.813 | 0.213 | - | - | - |
| | Disc-GS (Qu et al., 2024) | 28.01 | 0.833 | 0.189 | - | - | - |
| | DBS (Liu et al., 2025a) | 28.60 | 0.844 | 0.182 | 34.64 | 0.973 | 0.028 |
| | UBS-6D (**Ours**) | 28.66 | 0.840 | 0.184 | 34.92 | 0.975 | 0.026 |

Table 5 reports a quantitative comparison of UBS-7D against state-of-the-art kernel-based methods on the real-world N3V dataset (Li et al., 2022). UBS-7D achieves the best overall average PSNR of 32.22 dB, outperforming Ex4DGS (32.11 dB) and 7DGS (31.59 dB), while winning in 4 out of 6 scenes. This demonstrates the unified Beta primitive's effectiveness in capturing complex temporal dynamics and appearance variations in real-world dynamic content. Earlier methods 4DGS (30.19 dB) and 4DGaussians (28.63 dB) show significantly larger performance gaps, validating the advantage of our N-D anisotropic kernel design for challenging 4D reconstruction tasks.

Table 5: Comparison with other kernel methods on the real-world N3V dataset (Li et al., 2022).

| Model | Coffee Martini | Cook Spinach | Cut Roasted Beef | Flame Salmon | Flame Steak | Sear Steak | Average |
|---|---|---|---|---|---|---|---|
| 4DGS (Yang et al., 2024a) | 26.51 | 32.11 | 31.74 | 26.93 | 31.44 | 32.42 | 30.19 |
| 4DGaussians (Wu et al., 2024) | 26.69 | 31.89 | 25.88 | 27.54 | 28.07 | 31.73 | 28.63 |
| Ex4DGS (Lee et al., 2024b) | 28.79 | 33.23 | 33.73 | 29.29 | 33.91 | 33.69 | 32.11 |
| 7DGS (Gao et al., 2025b) | 29.07 | 32.23 | 32.93 | 28.83 | 33.07 | 33.41 | 31.59 |
| UBS-7D (**Ours**) | 29.73 | 33.30 | 33.88 | 28.77 | 33.83 | 33.83 | 32.22 |

### A.2  COMPARISONS ON THE 6DGS-PBR DATASET

Table 6 compares UBS-6D (Ours) with 3DGS (Kerbl et al., 2023), DBS (Liu et al., 2025a), and 6DGS (Gao et al., 2025a) on the 6DGS-PBR dataset. UBS-6D consistently achieves the best performance across all seven scenes, surpassing all baselines in reconstruction accuracy and perceptual quality. On average, UBS-6D achieves **40.10 dB** PSNR, **0.979** SSIM, and **0.039** LPIPS, outperforming

Table 6: Results on the 6DGS-PBR (Gao et al., 2025a) dataset. '*Train*' indicates time in minutes.

| Scene | 3DGS | | | DBS | | | 6DGS | | | UBS-6D (Ours) | | |
|---|---|---|---|---|---|---|---|---|---|---|---|---|
| | PSNR↑ | SSIM↑ | Train↓ | PSNR↑ | SSIM↑ | Train↓ | PSNR↑ | SSIM↑ | Train↓ | PSNR↑ | SSIM↑ | Train↓ |
| bunny | 29.19 | 0.985 | 22.8 | 30.10 | 0.982 | 23.5 | 37.41 | 0.991 | 24.1 | **45.68** | **0.996** | 32.1 |
| cloud | 34.03 | 0.985 | 23.1 | 43.88 | 0.993 | 12.7 | 41.00 | 0.991 | 21.6 | **46.81** | **0.995** | 30.6 |
| explosion | 27.06 | 0.953 | 11.3 | 32.42 | 0.965 | 17.9 | 41.61 | 0.990 | 18.3 | **44.54** | **0.994** | 32.4 |
| smoke | 26.82 | 0.964 | 24.2 | 28.85 | 0.964 | 47.5 | 41.41 | 0.993 | 31.1 | **44.02** | **0.995** | 33.5 |
| suzanne | 22.45 | 0.885 | 19.5 | 26.58 | 0.923 | 30.3 | 26.96 | 0.928 | 32.9 | **27.35** | **0.931** | 35.1 |
| dragon | 26.53 | 0.812 | 12.0 | 29.71 | 0.844 | 20.7 | 34.95 | 0.933 | 16.3 | **38.06** | **0.974** | 23.7 |
| ct-scan | 25.69 | 0.917 | 4.5 | 27.52 | 0.923 | 4.3 | 33.46 | 0.964 | 11.0 | **34.24** | **0.969** | 6.5 |
| avg | 27.40 | 0.929 | 16.8 | 31.29 | 0.942 | 22.41 | 36.68 | 0.970 | 22.2 | **40.10** | **0.979** | 27.7 |

Table 7: Per-primitive parameter count comparison across static and dynamic splatting methods. UBS significantly reduces parameter counts compared to existing NDGS methods: UBS-6D is 59% of 3DGS and UBS-7D is 27% of 4DGS.

| Comparison | Static Solutions | | | Dynamic Solutions | | |
|---|---|---|---|---|---|---|
| | 3DGS | 6DGS | UBS-6D (**Ours**) | 4DGS | 7DGS | UBS-7D (**Ours**) |
| Parameter (Geom. + Color) | $11 + 48 = 59$ | $28 + 48 = 76$ | $32 + 3 = 35$ | $17 + 144 = 161$ | $36 + 48 = 84$ | $41 + 3 = 44$ |
| Ratio | $1\times$ | $1.29\times$ | $0.59\times$ | $1\times$ | $0.52\times$ | $0.27\times$ |

6DGS by +3.4 dB PSNR and DBS by +8.8 dB, with comparable training efficiency (27.7 min vs. 22.2/22.4 min). The clear gap between 6DGS/UBS-6D and 3DGS/DBS arises from the presence of *conditional slicing*, which allows the mean and covariance to adapt with viewing conditions. In contrast, 3DGS and DBS rely on static spatial kernels, limiting their ability to capture these view-dependent effects. By combining high-dimensional Beta kernels with conditional slicing, UBS-6D attains complex visual effects and photorealism while maintaining competitive runtime.

# B    EFFICIENCY ANALYSIS

## B.1    PER-PRIMITIVE PARAMETER ANALYSIS

Table 7 provides a detailed breakdown of parameter counts per primitive across different splatting methods. UBS achieves significant parameter efficiency by eliminating the need for auxiliary color encodings through its native high-dimensional representation.

For static scenes, 3DGS requires 59 parameters per primitive: 11 for geometry (`pos`: 3, `scale`: 3, `rotation`: 4, `opacity`: 1) and 48 for spherical harmonic color encoding (16 coefficients $\times$ 3 RGB channels). 6DGS increases this to 76 parameters by expanding the covariance representation to 28 parameters while maintaining the same 48-parameter SH encoding. In contrast, UBS-6D requires only 35 parameters: 32 for $N$-dimensional geometry (`pos`: 3, `dir`: 3, `spatial-orthogonal covariance`: 21, `beta shapes`: 4, `opacity`: 1) and just 3 for direct RGB color, achieving a 41% parameter reduction compared to 3DGS.

For dynamic scenes, the parameter efficiency becomes even more pronounced. 4DGS requires 161 parameters per primitive: 17 for 4D geometry (`pos`: 3, `time`: 1, `4D-scale`: 4, `4D-rotation`: 8, `opacity`: 1) and 144 for temporal spherical harmonics (48 coefficients $\times$ 3 RGB channels). 7DGS reduces this to 84 parameters: 36 for geometry (`pos`: 3, `time`: 1, `dir`: 3, `7D covariance`: 28, `opacity`: 1) and 48 for standard SH encoding. UBS-7D achieves the most compact representation with only 44 parameters: 41 for unified 7D geometry (`pos`: 3, `time`: 1, `dir`: 3, `spatial-orthogonal covariance`: 28, `beta shapes`: 5, `opacity`: 1) and 3 for direct RGB color, representing a 73% parameter reduction compared to 4DGS.

This dramatic parameter efficiency stems from UBS's ability to natively encode view-dependent and temporal variations within the kernel itself, eliminating the need for separate spherical harmonic or temporal color encodings. The Beta kernel's inherent anisotropy across dimensions captures complex appearance variations that would otherwise require dozens of harmonic coefficients, making UBS both more expressive and more parameter-efficient than existing methods.

Table 8: Efficiency comparison with 3DGS and 6DGS on static benchmarks of NeRF Synthetic (Mildenhall et al., 2020), Mip-NeRF (Barron et al., 2022), 6DGS-PBR (Gao et al., 2025a), and with 4DGS and 7DGS on dynamic benchmarks of 7DGS-PBR (Gao et al., 2025b) and D-NeRF (Pumarola et al., 2021). 'Train' means training time in minutes; 'Mem' means storage size in MB.

| Dataset | Scene | 3DGS | | | | 6DGS | | | | UBS-6D (Ours) | | | |
|---|---|---|---|---|---|---|---|---|---|---|---|---|---|
| | | FPS↑ | Num↓ | Mem↓ | Train↓ | FPS↑ | Num↓ | Mem↓ | Train↓ | FPS↑ | Num↓ | Mem↓ | Train↓ |
| STATIC SCENES — NeRF Synthetic | chair | 412.8 | 489.0K | 115.7 | 4.8 | 420.4 | 410.7K | 119.1 | 19.6 | 356.3 | 300.0K | 40.1 | 8.7 |
| | drums | 562.3 | 390.9K | 92.5 | 4.1 | 555.8 | 317.2K | 92.0 | 15.4 | 349.1 | 300.0K | 40.1 | 8.1 |
| | ficus | 728.5 | 264.9K | 62.7 | 3.0 | 757.4 | 226.4K | 65.6 | 11.3 | 356.6 | 300.0K | 40.1 | 8.2 |
| | hotdog | 831.5 | 187.8K | 44.4 | 3.5 | 700.5 | 161.0K | 46.7 | 10.0 | 352.5 | 300.0K | 40.1 | 9.1 |
| | lego | 582.4 | 347.0K | 82.1 | 4.0 | 247.7 | 273.3K | 79.2 | 15.3 | 340.3 | 300.0K | 40.1 | 8.4 |
| | materials | 1286.1 | 160.1K | 37.9 | 2.7 | 1098.2 | 125.0K | 36.2 | 7.8 | 360.7 | 300.0K | 40.1 | 8.9 |
| | mic | 614.9 | 195.7K | 46.3 | 3.0 | 756.9 | 174.3K | 50.5 | 9.4 | 326.1 | 300.0K | 40.1 | 9.5 |
| | ship | 546.4 | 279.2K | 66.0 | 4.1 | 646.8 | 237.9K | 69.0 | 12.7 | 330.3 | 300.0K | 40.1 | 8.8 |
| | avg | 695.6 | 289.3K | 68.5 | 3.6 | 648.0 | 240.7K | 69.8 | 12.7 | 345.7 | 300.0K | 40.1 | 8.7 |
| Mip-NeRF 360 | bicycle | 105.2 | 6.0M | 1421.5 | 29.2 | 105.2 | 3.3M | 957.2 | 114.9 | 43.9 | 6.0M | 801.1 | 30.2 |
| | flowers | 150.9 | 3.6M | 850.0 | 19.6 | 150.9 | 2.4M | 707.2 | 88.8 | 84.3 | 3.0M | 400.5 | 19.6 |
| | garden | 84.2 | 5.7M | 1358.3 | 30.4 | 84.2 | 4.2M | 1219.4 | 164.9 | 52.9 | 5.0M | 667.6 | 36.8 |
| | stump | 151.2 | 4.6M | 1095.1 | 22.7 | 190.5 | 1.7M | 488.2 | 60.2 | 64.9 | 4.0M | 536.3 | 18.9 |
| | treehill | 161.6 | 3.8M | 886.5 | 20.1 | 130.9 | 2.6M | 755.8 | 94.0 | 72.4 | 3.5M | 467.3 | 21.9 |
| | room | 222.0 | 1.6M | 369.1 | 18.2 | 171.2 | 1.1M | 314.8 | 51.4 | 147.2 | 1.5M | 200.3 | 21.9 |
| | counter | 233.5 | 1.2M | 281.4 | 17.6 | 203.7 | 0.8M | 239.8 | 43.5 | 143.8 | 1.5M | 200.3 | 27.8 |
| | kitchen | 183.0 | 1.8M | 424.7 | 21.5 | 147.4 | 1.0M | 285.9 | 71.3 | 144.8 | 1.5M | 200.3 | 26.6 |
| | bonsai | 317.1 | 1.3M | 296.6 | 15.7 | 233.4 | 0.9M | 262.7 | 48.4 | 147.1 | 1.5M | 200.3 | 22.7 |
| | avg | 178.7 | 3.3M | 775.9 | 21.7 | 157.4 | 2.0M | 581.2 | 81.9 | 100.4 | 3.1M | 409.0 | 25.2 |
| 6DGS-PBR | bunny | 123.8 | 79.8K | 18.87 | 22.8 | 349.5 | 9.7K | 2.8 | 24.1 | 257.9 | 300.0K | 40.1 | 32.1 |
| | cloud | 370.5 | 60.0K | 14.19 | 23.1 | 429.8 | 17.9K | 5.2 | 21.6 | 247.7 | 300.0K | 40.1 | 30.6 |
| | explosion | 400.3 | 51.2K | 12.11 | 11.3 | 302.8 | 25.9K | 7.5 | 18.3 | 275.6 | 300.0K | 40.1 | 32.4 |
| | smoke | 176.6 | 55.8K | 13.20 | 24.2 | 207.7 | 21.0K | 6.1 | 31.1 | 225.0 | 300.0K | 40.1 | 33.5 |
| | suzanne | 186.5 | 316.2K | 74.78 | 19.5 | 94.9 | 208.3K | 60.4 | 32.9 | 325.9 | 300.0K | 40.1 | 35.1 |
| | dragon | 275.9 | 241.4K | 57.09 | 12.0 | 266.9 | 127.8K | 37.1 | 16.3 | 334.9 | 300.0K | 40.1 | 23.7 |
| | ct-scan | 607.2 | 229.2K | 54.20 | 4.5 | 726.8 | 179.3K | 52.0 | 11.0 | 370.2 | 300.0K | 40.1 | 6.5 |
| | avg | 305.9 | 147.7K | 34.9 | 16.8 | 339.8 | 84.3K | 24.4 | 22.2 | 290.2 | 300.0K | 40.1 | 27.7 |

| Dataset | Scene | 4DGS | | | | 7DGS | | | | UBS-7D (Ours) | | | |
|---|---|---|---|---|---|---|---|---|---|---|---|---|---|
| | | FPS↑ | Num↓ | Mem↓ | Train↓ | FPS↑ | Num↓ | Mem↓ | Train↓ | FPS↑ | Num↓ | Mem↓ | Train↓ |
| DYNAMIC SCENES — 7DGS-PBR | heart1 | 186.9 | 694.0K | 444.6 | 103.0 | 401.0 | 82.4K | 27.7 | 114.2 | 244.6 | 328.0K | 55.1 | 38.7 |
| | heart2 | 160.4 | 869.2K | 556.9 | 103.4 | 384.6 | 101.5K | 34.1 | 384.6 | 297.1 | 328.0K | 55.1 | 33.5 |
| | cloud | 219.0 | 216.9K | 138.9 | 123.7 | 386.2 | 44.2K | 14.8 | 102.6 | 473.6 | 150.0K | 25.2 | 50.0 |
| | dust | 296.1 | 357.7K | 229.2 | 97.0 | 394.8 | 10.9K | 3.7 | 69.8 | 406.0 | 150.0K | 25.2 | 46.5 |
| | flame | 151.2 | 947.8K | 607.2 | 113.7 | 371.6 | 15.1K | 5.1 | 74.1 | 461.4 | 150.0K | 25.2 | 53.7 |
| | suzanne | 141.8 | 766.1K | 1,701.6 | 222.5 | 317.9 | 276.3K | 92.8 | 193.9 | 275.0 | 300.0K | 50.4 | 136.8 |
| | avg | 192.6 | 642.0K | 613.1 | 127.2 | 376.0 | 88.4K | 29.7 | 116.7 | 359.6 | 234.3K | 39.4 | 59.9 |
| D-NeRF | b.balls | 219.9 | 276.1K | 277.7 | 50.7 | 213.1 | 127.4K | 43.5 | 86.6 | 366.6 | 150.0K | 25.2 | 49.6 |
| | h.warrior | 299.4 | 298.4K | 212.6 | 35.3 | 431.5 | 8.7K | 2.9 | 31.3 | 395.2 | 150.0K | 25.2 | 36.3 |
| | hook | 325.1 | 174.7K | 183.3 | 38.0 | 432.8 | 21.7K | 7.3 | 35.7 | 452.2 | 150.0K | 25.2 | 34.0 |
| | j.jacks | 366.0 | 143.7K | 534.8 | 66.9 | 432.8 | 21.7K | 5.3 | 34.1 | 376.4 | 150.0K | 25.2 | 43.2 |
| | lego | 320.0 | 186.2K | 355.0 | 55.4 | 365.0 | 68.6K | 31.4 | 78.5 | 347.4 | 150.0K | 25.2 | 31.3 |
| | mutant | 341.6 | 138.7K | 88.9 | 39.1 | 395.8 | 33.9K | 11.5 | 42.5 | 258.7 | 300.0K | 50.4 | 60.0 |
| | standup | 330.4 | 142.5K | 107.1 | 34.4 | 399.2 | 12.7K | 4.3 | 33.5 | 389.4 | 150.0K | 25.2 | 41.1 |
| | trex | 169.2 | 682.4K | 1,085.2 | 100.7 | 352.4 | 62.0K | 21.4 | 63.4 | 359.4 | 150.0K | 25.2 | 41.6 |
| | avg | 296.4 | 255.3K | 380.3 | 52.6 | 377.8 | 43.8K | 16.0 | 50.7 | 368.2 | 168.8K | 28.4 | 42.1 |

Table 9: Impact of primitive capacity on NeRF Synthetic dataset. 'Mem' means storage size in MB.

| Scene | 100K Primitives | | | | 200K Primitives | | | | 300K Primitives | | | |
|---|---|---|---|---|---|---|---|---|---|---|---|---|
| | PSNR↑ | SSIM↑ | Mem↓ | FPS↑ | PSNR↑ | SSIM↑ | Mem↓ | FPS↑ | PSNR↑ | SSIM↑ | Mem↓ | FPS↑ |
| chair | 36.16 | 0.989 | 13.4 | 543.4 | 36.62 | 0.990 | 26.7 | 341.3 | 36.62 | 0.990 | 40.1 | 356.3 |
| drums | 26.88 | 0.959 | 13.4 | 529.3 | 27.01 | 0.959 | 26.7 | 376.3 | 27.03 | 0.959 | 40.1 | 349.1 |
| ficus | 36.17 | 0.988 | 13.4 | 574.4 | 36.44 | 0.989 | 26.7 | 431.9 | 36.57 | 0.989 | 40.1 | 356.6 |
| hotdog | 38.68 | 0.988 | 13.4 | 534.7 | 38.81 | 0.988 | 26.7 | 366.9 | 38.94 | 0.988 | 40.1 | 352.5 |
| lego | 35.66 | 0.981 | 13.4 | 572.1 | 36.67 | 0.984 | 26.7 | 370.7 | 36.97 | 0.985 | 40.1 | 340.3 |
| materials | 32.49 | 0.974 | 13.4 | 531.7 | 32.85 | 0.976 | 26.7 | 367.7 | 32.93 | 0.977 | 40.1 | 360.7 |
| mic | 37.48 | 0.994 | 13.4 | 493.1 | 37.79 | 0.994 | 26.7 | 346.7 | 37.92 | 0.994 | 40.1 | 326.1 |
| ship | 31.88 | 0.911 | 13.4 | 543.4 | 32.10 | 0.913 | 26.7 | 344.8 | 32.09 | 0.913 | 40.1 | 330.3 |
| avg | 34.42 | 0.973 | 13.4 | 540.3 | 34.79 | 0.974 | 26.7 | 368.3 | 34.89 | 0.974 | 40.1 | 345.7 |

Table 10: Comparison of rotation parameterizations for covariance matrix construction on the `bicycle` scene for UBS-3D. Our spatial-orthogonal approach balances performance, efficiency, and parameter count.

| Covariance Matrix Parameterization | Training Time | PSNR | # Parameters (N>3) | # Parameters (N=3) |
|---|---|---|---|---|
| Quaternion (w/o CUDA) | 3h26m | 25.50 | Not Applicable | 7 |
| Cholesky | 1h14m | 25.28 | N(1+N)/2 | 6 |
| Skew-Symmetric w/ Matrix Exp | 3h6m | 25.49 | N(1+N)/2 | 6 |
| Skew-Symmetric w/ Cayley Transform | 1h16m | 25.52 | N(1+N)/2 | 6 |
| Skew-Symmetric w/ 2nd order Taylor | 1h27m | **25.54** | N(1+N)/2 | 6 |
| Skew-Symmetric w/ 1st order Taylor | **1h** | **25.54** | N(1+N)/2 | 6 |

## B.2 COMPUTATIONAL EFFICIENCY

Table 8 presents a comprehensive efficiency comparison between UBS and baseline methods. For static scenes, UBS-6D achieves 31% faster training than 6DGS on NeRF Synthetic (8.7 vs 12.7 minutes) with 43% memory reduction (40.1 vs 69.8 MB) and 345.7 FPS rendering. On Mip-NeRF 360, UBS achieves the lowest memory usage (409.0 MB, 47% and 30% reduction vs 3DGS and 6DGS) with 100.4 FPS. On 6DGS-PBR, UBS achieves comparable training times (27.7 vs 22.2 minutes for 6DGS) with 290.2 FPS. For dynamic scenes, UBS-7D reduces training time by 53% and 49% compared to 4DGS and 7DGS on 7DGS-PBR (59.9 vs 127.2 and 116.7 minutes) with 359.6 FPS, and by 20% and 17% on D-NeRF (42.1 vs 52.6 and 50.7 minutes) with 368.2 FPS. The efficiency gains stem from UBS's unified Beta kernel design, which eliminates auxiliary spherical harmonic encodings. Moreover, unlike previous Gaussian methods that rely on heuristic densification and pruning, UBS employs MCMC-based optimization that allows direct control over primitive count, enabling flexible speed-quality trade-offs as demonstrated in Table 9.

Table 9 demonstrates this flexibility by analyzing the speed-quality trade-off across varying primitive capacities (100K, 200K, 300K) on NeRF Synthetic. Results reveal that reducing primitives from 300K to 100K decreases quality by only 0.47 dB PSNR (34.89 to 34.42 dB) while significantly improving rendering speed by 56% (345.7 to 540.3 FPS) and reducing memory by 67% (40.1 to 13.4 MB). This demonstrates that UBS can be optimized for speed-critical applications by adaptively adjusting primitive counts to balance quality and efficiency.

## C SPATIAL-ORTHOGONAL CHOLESKY PARAMETERIZATION

Our spatial-orthogonal Cholesky parameterization (Equation 6 in the main paper) addresses a fundamental challenge in $N$-dimensional radiance field rendering: how to parameterize covariance matrices that simultaneously preserve interpretable 3D spatial structure while enabling flexible cross-dimensional correlations. This section provides theoretical justification for our design and empirical validation of our rotation parameterization choice.

### C.1 THEORETICAL ANALYSIS

**Design Requirements.** For the covariance matrix $\boldsymbol{\Sigma} \in \mathbb{R}^{N \times N}$ in $N$-dimensional primitives, we require a parameterization that: (1) guarantees positive semi-definiteness (PSD), (2) preserves 3D spatial interpretability (rotation + scale structure), (3) enables cross-dimensional correlations with non-spatial dimensions (view, time), (4) generalizes to arbitrary $N$ dimensions, and (5) computes efficiently during optimization.

Existing approaches fall short of these requirements. Quaternion-based rotation parameterization (used in 3DGS) provides excellent spatial interpretability but does not generalize beyond 3D. Pure Cholesky factorization generalizes to arbitrary dimensions but loses explicit spatial rotation structure, leading to suboptimal geometric representation (Table 10 shows 0.22 dB PSNR drop). Naive matrix exponential approaches preserve orthogonality and generalize to $N$-D but incur prohibitive computational costs (3h+ training time).

**Construction.** We construct the covariance through a hybrid Cholesky factorization $\boldsymbol{\Sigma} = \boldsymbol{L}\boldsymbol{L}^\top$ with structured lower-triangular matrix:

$$\boldsymbol{L} = \begin{pmatrix} \boldsymbol{R}_x \text{diag}(\boldsymbol{s}_x) & \boldsymbol{0} \\ \boldsymbol{L}_{qx} & \boldsymbol{L}_q \end{pmatrix}, \tag{14}$$

where $\boldsymbol{R}_x \in SO(3)$ is the spatial rotation matrix, $\boldsymbol{s}_x \in \mathbb{R}^3$ are spatial scales, $\boldsymbol{s}_q \in \mathbb{R}^C$ are non-spatial scales for $C$ additional dimensions, and $\boldsymbol{L}_{qx} \in \mathbb{R}^{C \times 3}$ encodes cross-dimensional correlations. This yields:

$$\boldsymbol{\Sigma} = \begin{pmatrix} \boldsymbol{R}_x \text{diag}(\boldsymbol{s}_x)^2 \boldsymbol{R}_x^\top & \boldsymbol{R}_x \text{diag}(\boldsymbol{s}_x)\boldsymbol{L}_{qx}^\top \\ \boldsymbol{L}_{qx}\text{diag}(\boldsymbol{s}_x)\boldsymbol{R}_x^\top & \boldsymbol{L}_{qx}\boldsymbol{L}_{qx}^\top + \boldsymbol{L}_q\boldsymbol{L}_q^\top \end{pmatrix}. \tag{15}$$

**Skew-Symmetric Rotation Parameterization.** For the spatial rotation matrix $\boldsymbol{R}_x \in SO(3)$, we employ a skew-symmetric matrix parameterization. A skew-symmetric matrix $\boldsymbol{A}_x \in \mathbb{R}^{3 \times 3}$ satisfies $\boldsymbol{A}_x = -\boldsymbol{A}_x^\top$, which in 3D has the explicit form:

$$\boldsymbol{A}_x = \begin{pmatrix} 0 & -\omega_3 & \omega_2 \\ \omega_3 & 0 & -\omega_1 \\ -\omega_2 & \omega_1 & 0 \end{pmatrix}, \quad \boldsymbol{\omega} = (\omega_1, \omega_2, \omega_3)^\top \in \mathbb{R}^3, \tag{16}$$

where $\boldsymbol{\omega}$ represents the learnable rotation parameters. The rotation matrix is obtained via the matrix exponential map:

$$\boldsymbol{R}_x = \exp(\boldsymbol{A}_x) = \sum_{k=0}^{\infty} \frac{\boldsymbol{A}_x^k}{k!} = \boldsymbol{I} + \boldsymbol{A}_x + \frac{\boldsymbol{A}_x^2}{2!} + \frac{\boldsymbol{A}_x^3}{3!} + \cdots, \tag{17}$$

which mathematically guarantees orthogonality ($\boldsymbol{R}_x^\top \boldsymbol{R}_x = \boldsymbol{I}$) and unit determinant ($\det(\boldsymbol{R}_x) = 1$), ensuring $\boldsymbol{R}_x \in SO(3)$. For computational efficiency, we use the first-order Taylor approximation:

$$\boldsymbol{R}_x \approx \boldsymbol{I} + \boldsymbol{A}_x, \tag{18}$$

which maintains approximate orthogonality for small $\|\boldsymbol{\omega}\|$ while avoiding the expensive matrix exponential computation. This approximation initializes with $\boldsymbol{\omega} = \boldsymbol{0}$ (identity rotation) and progressively learns rotation during optimization.

**Mathematical Properties.** Our parameterization satisfies all design requirements: **(1) PSD:** Guaranteed by Cholesky properties ($\boldsymbol{\Sigma} = \boldsymbol{L}\boldsymbol{L}^\top$). **(2) Spatial Interpretability:** The spatial block $\boldsymbol{\Sigma}_x = \boldsymbol{R}_x \text{diag}(\boldsymbol{s}_x)^2 \boldsymbol{R}_x^\top$ exactly recovers 3DGS parameterization. **(3) Cross-Correlations:** Off-diagonal block $\boldsymbol{\Sigma}_{xq} = \boldsymbol{R}_x \text{diag}(\boldsymbol{s}_x)\boldsymbol{L}_{qx}^\top$ enables view-dependent spatial deformation. **(4) Generalization:** Holds for arbitrary $N = 3 + C$ dimensions. **(5) Efficiency:** Spatial rotation $\boldsymbol{R}_x$ uses skew-symmetric first-order Taylor approximation $\boldsymbol{R}_x \approx \boldsymbol{I} + \boldsymbol{A}_x$, avoiding expensive matrix exponential while maintaining approximate orthogonality.

### C.2 EMPIRICAL ANALYSIS

Having established the structure, we now validate our choice of spatial rotation parameterization through systematic comparison. We evaluate six strategies on the `bicycle` scene from Mip-NeRF 360 using UBS-3D (pure spatial Beta splatting), focusing on the 3D spatial rotation subspace.

Table 10 presents our findings across different parameterization strategies:

**Quaternion.** The traditional 3DGS approach using quaternion rotation and diagonal scaling achieves competitive PSNR (25.50 dB) and the lowest parameter count (7 parameters: 3 position + 3 scale + 1 quaternion magnitude after normalization). However, this approach does not generalize to higher dimensions ($N > 3$), making it unsuitable for our $N$-dimensional framework.

**Cholesky Decomposition.** Pure Cholesky factorization $\boldsymbol{\Sigma} = \boldsymbol{L}\boldsymbol{L}^\top$ offers faster training (1h14m) and generalizes to arbitrary dimensions with $N(N+1)/2$ parameters. However, it suffers from a 0.22 dB PSNR drop compared to quaternions, as it lacks explicit orthogonality constraints that help maintain geometric consistency in the spatial subspace.

**Skew-Symmetric Matrix Methods.** Constructing spatial rotation matrices through $\boldsymbol{R} = \exp(\boldsymbol{A})$ where $\boldsymbol{A}$ is skew-symmetric preserves orthogonality and generalizes to higher dimensions. The naive

implementation requiring full matrix exponentiation is prohibitively slow (3h6m), but approximation methods offer significant improvements. The Cayley transform $\boldsymbol{R} = (\boldsymbol{I} - \boldsymbol{A})^{-1}(\boldsymbol{I} + \boldsymbol{A})$ reduces training to 1h16m with 25.52 dB PSNR. Taylor approximations provide the best balance: the 1st order Taylor approximation $\boldsymbol{R} \approx \boldsymbol{I} + \boldsymbol{A}$ achieves both the fastest training time (1h) and the highest PSNR (25.54 dB), making it optimal for spatial rotation parameterization.

**Spatial-Orthogonal Cholesky.** Based on these results, we select the skew-symmetric approach with 1st order Taylor approximation for spatial rotation parameterization, as it provides the best combination of training speed (1h) and reconstruction quality (25.54 dB). For our full UBS framework, we adopt a hybrid spatial-orthogonal approach: skew-symmetric 1st order Taylor for the 3D spatial rotation subspace (preserving geometric interpretability and optimal performance) combined with Cholesky factors for modeling cross-dimensional correlations between spatial and non-spatial dimensions (view, time). This design achieves the optimal trade-off between performance, generalizability, and computational efficiency across all UBS variants.

The analysis confirms that the skew-symmetric 1st order Taylor approximation provides the best spatial rotation parameterization, enabling our spatial-orthogonal design to maintain geometric consistency in the spatial subspace while allowing flexible correlations with additional dimensions in the full $N$-dimensional UBS framework.

# D    THEORETICAL FOUNDATION AND ANALYSIS

**Fundamental Limitations of Gaussian Kernels**    $N$-dimensional Gaussian kernels suffer from inherent dimensional coupling that prevents modeling of independent scene properties. In methods like 6DGS and 7DGS, all dimensions are coupled through the joint Gaussian distribution: when the input view direction or time changes, the primitive is inevitably affected across all dimensions—it shifts in space, changes shape, and modulates opacity simultaneously. This coupling stems from two fundamental constraints of the Gaussian kernel:

**1. Fixed Bell-Shaped Profile:** The Gaussian kernel maintains its characteristic bell-shaped exponential decay $\exp(-r^2/2\sigma^2)$ across all dimensions. This fixed functional form cannot produce flat responses needed for view-independent or temporally-static elements. For example, in 6DGS, a primitive cannot remain view-independent because the Gaussian's bell-shaped profile in the angular dimensions will always modulate opacity based on viewing direction, even when the scene element should appear constant across views.

**2. Symmetric Dimensional Coupling:** The multivariate Gaussian distribution enforces symmetric relationships between dimensions through its covariance structure. When conditioned on viewing direction or time, the resulting 3D Gaussian inevitably shifts its spatial mean, changes its spatial covariance, and modulates its opacity according to the angular or temporal distance from the mean. This prevents primitives from exhibiting independence—a spatially-localized texture cannot remain fixed while only its angular response varies, or a static object cannot maintain constant appearance while dynamic elements change around it.

These limitations force inefficient representations where many coupled primitives must approximate what should be expressible by fewer independent primitives.

In contrast, UBS's Beta kernels enable per-dimension shape control: spatial dimensions can adopt flat profiles for view-independent geometry while angular dimensions use peaked responses for specular highlights, and temporal dimensions can remain nearly constant for static elements while varying sharply for dynamic components. This dimensional independence allows each primitive to model complex anisotropic phenomena efficiently—a single UBS primitive can simultaneously represent a static spatial surface with view-dependent reflectance, whereas Gaussian methods require multiple coupled primitives that cannot cleanly separate these properties.

**Universal Beta Kernels: Mathematical Properties.**    UBS addresses these limitations through the $N$-dimensional Beta kernel:

$$\mathcal{B}(\boldsymbol{x}; \boldsymbol{\mu}, \boldsymbol{\Sigma}, \boldsymbol{b}) = \prod_{i=1}^{N} (1 - d_i(\boldsymbol{x}))^{\beta_i}, \quad d_i \in [0, 1), \tag{19}$$

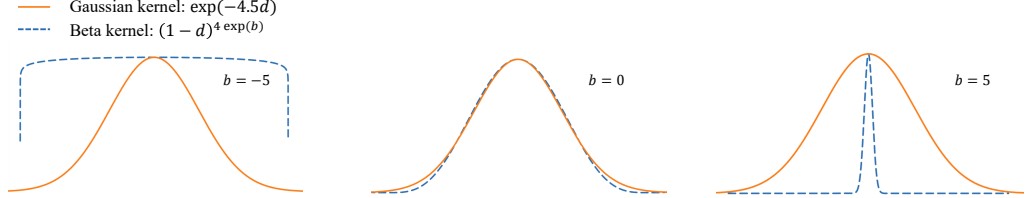

Figure 4: Adaptive Beta kernels defined on $d \in [0, 1)$. The curves are extended symmetrically to $(-1, 1)$ for illustrative visualization only.

where $d_i(\boldsymbol{x})$ represents the bounded distance. For the first three dimensions ($i = 1, 2, 3$), $d_i$ is defined by $r_i$ in Equation 1, while for the remaining dimensions ($i = 4, \ldots, N$), $d_i$ is defined by Equation 11. The shape parameter $\beta_i = 4 \exp(b_i)$ controls the kernel profile.

The Beta kernel exhibits three key properties:

**Property 1 (Shape Adaptivity):** The shape parameter $b_i$ directly controls kernel behavior through $\beta_i = 4 \exp(b_i)$:

- $b_i < 0$: Flatter kernel profile ($\beta_i < 4$, broader support)
- $b_i = 0$: Gaussian-like profile ($\beta_i = 4$, standard bell curve)
- $b_i > 0$: More peaked kernel profile ($\beta_i > 4$, sharper localization)

When $b_i = -5$, we have $\beta_i = 4e^{-5} \approx 0.027$ and the kernel becomes nearly constant $(1-d_i)^{0.027} \approx 1$. When $b_i = 5$, $\beta_i = 4e^5 \approx 594$ and the kernel becomes highly peaked. These practical ranges ($b_i \in [-5, 5]$) are sufficient to span from flat to sharp kernel profiles, as shown in Figure 4.

**Property 2 (Per-Dimension Shape Control):** The gated product form enables independent shape control for each dimension through its own $b_i$:

$$\mathcal{B}(\boldsymbol{x}) = \prod_{i=1}^{N} (1 - d_i(\boldsymbol{x}))^{4 \exp(b_i)}, \tag{20}$$

Unlike Gaussian kernels where all dimensions share the same bell-shaped functional form (only varying in scale), Beta kernels allow each dimension to adopt fundamentally different shapes: dimension $i$ can be nearly flat ($b_i = -3$) while dimension $j$ is highly peaked ($b_j = 3$). This per-dimension shape control, combined with cross-dimensional correlations through $\boldsymbol{\Sigma}$, enables efficient representation of complex anisotropic phenomena.

**Property 3 (Backward Compatibility):** When $b_i = 0$, we have $\beta_i = 4 \exp(0) = 4$, and the Beta kernel approximates a Gaussian:

$$(1 - d)^4 \approx \exp(-4.5d^2) \quad \text{for } d \in [0, 1), \tag{21}$$

guaranteeing that UBS subsumes Gaussian methods as special cases when $\boldsymbol{b} = \boldsymbol{0}$. This ensures seamless compatibility with existing Gaussian-based methods.

