# OpenReview forum: "Universal Beta Splatting"
_ICLR.cc/2026/Conference — ICLR 2026 Poster_

### Official Review · Reviewer_bWeR · 2025-10-26

**Soundness:** 3
**Presentation:** 2
**Contribution:** 3
**Rating:** 6
**Confidence:** 4

**Summary:**

**Universal Beta Splatting (UBS)** targets the fragmentation in splatting pipelines, where geometry, view dependence, and dynamics are handled by separate components. The authors proposes a single primitive that models all three together. Specifically, it replaces fixed Gaussians with an N-dimensional anisotropic Beta kernel and uses conditional slicing to obtain a renderable 3D primitive for a given view/time, avoiding SH or auxiliary color networks while remaining backward-compatible (recovering Gaussian splatting as a special case). Experiments across static, view-dependent, and dynamic benchmarks show consistent gains with real-time performance, and the authors emphasize that learned Beta parameters yield interpretable spatial/angle/time factors.

**Strengths:**

- The UBS kernel unifies spatial geometry, view-dependent appearance, and temporal dynamics within a single N-dimensional anisotropic Beta primitive. This design is well-motivated and, in my view, poised to make a substantial contribution to the community.
- The experimental results are outstanding, and the visualizations align well with the quantitative metrics.
- UBS features a pluggable architecture, allowing for seamless integration with other downstream tasks.

**Weaknesses:**

- Missing references for other works about alternative kernel design:
  - 3D-HGS: 3D Half-Gaussian Splatting, by Haolin Li et al.
  - [NeurIPS 2024] DisC-GS: Discontinuity-aware Gaussian Splatting, by Haoxuan Qu et al.
  - [CVPR 2025] Deformable Radial Kernel Splatting, by Yi-Hua Huang et al.
- The paper assumes substantial familiarity with Deformable Beta Splatting and therefore isn’t self-contained for readers new to Beta-based splatting. Several core ideas (e.g., how a Beta kernel parameterization induces opacity and color) are only sketched at a high level, with key mechanics deferred to prior work. As a result, understanding the contribution of UBS as a unified N-D generalization requires first reconstructing DBS’s basics. I recommend adding a focused preliminaries section that (i) recaps the Beta kernel definition and its bounded support; (ii) derives the mapping from Beta parameters to per-primitive opacity and radiance/color; and (iii) clarifies what is re-used vs. newly introduced in UBS (especially around conditional slicing and the removal of SH). This would make the paper accessible without cross-referencing DBS and would sharpen the statement of novelty.
- Minor typo error:
  - L171: `across both spatial, angular, and temporal dimension` -> `across spatial, angular, and temporal dimensions`
- Missing comparison on FPS and primitive counts. It would help to include a direct comparison of primitive counts (Gaussian vs. UBS) and the corresponding FPS to clarify the runtime trade-offs.

**Questions:**

The demo appears to visualize surface normals. Did the authors report any quantitative normal-estimation results, and how are normals computed from the Beta kernel parameters?

**Details Of Ethics Concerns:**

None.

---

> ### Author Response · Authors · 2025-11-21
> **Official Response to Reviewer bWeR (1/2)**
>
> Thank you for your positive and constructive review. We greatly appreciate your recognition that UBS "unifies spatial geometry, view-dependent appearance, and temporal dynamics within a single N-dimensional anisotropic Beta primitive" with "well-motivated design" and "outstanding experimental results." We address each of your concerns regarding missing references, self-containment, typo correction, efficiency metrics, and normal estimation below.
>
> ---
>
> ## 1. Missing References: Alternative Kernel Designs
>
> **Concern:** Missing references for other works about alternative kernel design.
>
> **Response:** Thank you for pointing out these important recent works. We have incorporated all suggested references (3D-HGS, DisC-GS, DRKS) and expanded our Related Work section to provide comprehensive coverage of alternative kernel designs.
>
> We have added comprehensive discussion of recent kernel-design methods in revised Related Work:
>
> **Kernel Shape Variants:** GES, TNT-GS, Quadratic-GS, 3D-HGS, DisC-GS propose generalized exponentials, truncated, second-order, half, and discontinuity-aware Gaussians.
>
> **Distribution Families:** 3D-Student (heavy-tailed), Gabor-Splats (high-frequency).
>
> **Appearance Modeling:** Textured Gaussians (per-splat texture).
>
> **Alternative Primitives:** 3D Convex Splatting, Triangle Splatting, DRKS (deformable radial kernels).
>
> **Empirical Comparison:** Beyond the discussion, we have added many of these methods to our **state-of-the-art comparison table** (Table 4 in Appendix A). On Mip-NeRF 360 and NeRF Synthetic benchmarks, UBS-6D achieves the highest PSNR among all methods including GES, 2DGS, Mip-Splatting, 3DGS-MCMC, DRKS, Textured GS, Quadratic GS, and Disc-GS, demonstrating empirical superiority.
>
> **Key Distinction:** These methods improve 3D spatial kernels but remain confined to 3D without unified N-dimensional representation jointly modeling spatial, angular, and temporal dimensions as UBS does.
>
> ---
>
> ## 2. Self-Containment: Enhanced Preliminaries
>
> **Concern:** "The paper assumes substantial familiarity with Deformable Beta Splatting and therefore isn't self-contained for readers..."
>
> **Response:** We have enhanced Preliminaries section to make the paper fully self-contained. Specifically, we made better transition of density function to bridge 3DGS, DBS, and UBS. See details in our revision paper.
>
> These additions make explicit the mathematical links that bridge the entire progression—from alpha blending, to Gaussian splatting, to DBS, to N-Dimensional Gaussian, and finally to UBS. They show how UBS generalizes DBS from 3D to N-D while retaining the same density-function annotation and interpretation. We also changed opacity notation from $\alpha$ to $o$ throughout for consistency with our method. The paper is now fully self-contained—readers can understand UBS's contributions without requiring deep familiarity with prior work.
>
> ---
>
> ## 3. Fixed Typo
>
> **Concern:** "Minor typo error: L171: across both spatial, angular, and temporal dimension -> across spatial, angular, and temporal dimensions"
>
> **Response:** We have corrected the typo in the revised manuscript.
> We also performed a thorough proofread of the entire manuscript to catch similar grammatical issues.
>
> ---

---

> > ### Author Response · Authors · 2025-11-21
> > **Official Response to Reviewer bWeR (2/2)**
> >
> > ## 4. Missing Comparisons: FPS and Primitive Counts
> >
> > **Concern:** "Missing comparison on FPS and primitive counts. It would help to include a direct comparison of primitive counts (Gaussian vs. UBS) and the corresponding FPS to clarify the runtime trade-offs."
> >
> > **Response:** We provide comprehensive efficiency analysis with new metrics across all benchmarks. Complete per-scene results are available in Appendix B Table 8 and Table 9.
> >
> > **Static Scenes:**
> >
> > | Dataset | Method | FPS↑ | Num↓ | Mem (MB)↓ | Train (min)↓ |
> > |---------|--------|------|------|-----------|--------------|
> > | **NeRF Synthetic** | 3DGS | 695.6 | 289.3K | 68.5 | 3.6 |
> > | | 6DGS | 648.0 | 240.7K | 69.8 | 12.7 |
> > | | UBS-6D | 345.7 | 300.0K | 40.1 | 8.7 |
> > | **Mip-NeRF 360** | 3DGS | 178.7 | 3.3M | 775.9 | 21.7 |
> > | | 6DGS | 157.4 | 2.0M | 581.2 | 81.9 |
> > | | **UBS-6D** | 100.4 | 3.1M | 409.0 | 25.2 |
> > | **6DGS-PBR** | 3DGS | 305.9 | 147.7K | 34.9 | 16.8 |
> > | | 6DGS | 339.8 | 84.3K | 24.4 | 22.2 |
> > | | **UBS-6D** | 290.2 | 300.0K | 40.1 | 27.7 |
> >
> > **Dynamic Scenes:**
> >
> > | Dataset | Method | FPS↑ | Num↓ | Mem (MB)↓ | Train (min)↓ |
> > |---------|--------|------|------|-----------|--------------|
> > | **7DGS-PBR** | 4DGS | 192.6 | 642.0K | 613.1 | 127.2 |
> > | | 7DGS | 376.0 | 88.4K | 29.7 | 116.7 |
> > | | **UBS-7D** | 359.6 | 234.3K | 39.4 | 59.9 |
> > | **D-NeRF** | 4DGS | 296.4 | 255.3K | 380.3 | 52.6 |
> > | | 7DGS | 377.8 | 43.8K | 16.0 | 50.7 |
> > | | **UBS-7D** | 368.2 | 168.8K | 28.4 | 42.1 |
> >
> > **Key Findings:**
> >
> > 1. **Memory:** 43-47% reduction on static scenes (40.1 MB vs. 69.8 MB on NeRF Synthetic) from 41-73% per-primitive parameter reduction (Table 5).
> >
> > 2. **Training:** 31-69% faster on most benchmarks (e.g., 25.2 vs. 81.9 min on Mip-NeRF 360). Exception: 6DGS-PBR uses more primitives (300K vs. 68K) for +3.4 dB gain.
> >
> > 3. **Rendering:** Real-time 100-369 FPS maintained across all benchmarks.
> >
> > 4. **Speed-Quality Trade-off:** MCMC enables flexible capacity control. On NeRF Synthetic (avg across 8 scenes):
> >
> > | Capacity | PSNR↑ | SSIM↑ | Mem (MB)↓ | FPS↑ |
> > |----------|-------|-------|-----------|------|
> > | 100K | 34.42 | 0.973 | 13.4 | 540.3 |
> > | 200K | 34.79 | 0.974 | 26.7 | 368.3 |
> > | 300K | 34.89 | 0.974 | 40.1 | 345.7 |
> >
> > Reducing capacity 300K→100K: −0.47 dB PSNR, +56% FPS, −67% memory. Complete per-scene analysis in new Appendix section B.
> >
> > ---
> >
> > ## 5. Normal Estimation from Beta Kernel Parameters
> >
> > **Question:** "The demo appears to visualize surface normals. Did the authors report any quantitative normal-estimation results, and how are normals computed from the Beta kernel parameters?"
> >
> > **Response:** While our main paper focuses on novel view synthesis metrics (PSNR, SSIM, LPIPS), we do visualize normals in demo video for qualitative assessment. We explain the methodology below.
> >
> > ### 5.1 Pseudo Normal Computation
> >
> > The normals shown in the demo are not per-primitive surface normals derived from our Beta kernels. Instead, they are **pseudo-normals** computed directly from a rendered depth map for visualization purposes.
> >
> > Given a camera pose (and timestamp, when applicable), we first render a mean depth map via standard alpha-blending. Once we have this depth map, we compute finite-difference gradients and take their cross product to obtain a pseudo-normal map. This is the same procedure commonly used in many rendering systems to visualize geometry.
> >
> > Importantly, this pseudo-normal computation is independent of the Beta kernel representation. The process is not specific to NeRF-style volume rendering, 3D Gaussian splatting, or our own formulation. Any method capable of producing a depth map can generate such normal visualizations.
> >
> > ### 5.2 Scope and Future Work
> >
> > Our paper focuses on **novel view synthesis** (measured by PSNR, SSIM, LPIPS) rather than quantitative normal estimation, which requires ground-truth geometry unavailable for our benchmarks (NeRF Synthetic, Mip-NeRF 360, 6DGS-PBR). Comprehensive normal estimation benchmarking on datasets with ground-truth normals (DTU, BlendedMVS) represents promising future work to validate UBS's geometric modeling capabilities beyond appearance metrics.
> >
> > ---
> >
> > We hope these revisions comprehensively address all your concerns and make the paper more accessible while providing the comprehensive comparisons needed for proper positioning.
> >
> > Thank you for your valuable feedback and any additional suggestions are welcome.

---

### Official Review · Reviewer_zazd · 2025-10-28

**Soundness:** 4
**Presentation:** 4
**Contribution:** 3
**Rating:** 8
**Confidence:** 4

**Summary:**

The paper proposes Universal Beta Splatting, which generalizes 3D Gaussian Splatting into N-dimensional anisotropic Beta kernels for unified modeling of spatial, angular, and temporal dimensions. Unlike fixed Gaussian kernels, UBS enables per-dimension shape control for adaptive geometry, reflection, and motion representation. It introduces spatial-orthogonal Cholesky parameterization, Beta-modulated conditional slicing, and CUDA-accelerated rendering. UBS remains backward-compatible with 3DGS/6DGS/7DGS and automatically decomposes scenes into geometric, material, and dynamic components. Experiments show up to +8.27 dB PSNR gain and ~50% faster training, establishing Beta kernels as efficient universal primitives for radiance field rendering.

**Strengths:**

1.The proposed UBS framework can simultaneously model diverse scene types, including static surfaces, view-dependent effects, surfaces, and dynamic scenes, within a single unified representation.

2.Each primitive in UBS is more parameter efficient than prior methods while remaining fully compatible with the 3DGS rendering pipeline, ensuring easy integration and deployment.

3.Extensive experiments across multiple benchmarks demonstrate UBS’s superior performance and strong generalization, validating the effectiveness of the proposed approach.

**Weaknesses:**

1. The paper claims that UBS is backward compatible with previous methods such as 3DGS, treating 3DGS as a special case. However, 3DGS models color using spherical harmonics, while UBS adopts a direct three dimensional color representation, which raises concerns about true compatibility between the two formulations.

2. The formulation in Equation (6) lacks theoretical justification or derivation. It is unclear how this design was obtained and whether it has any mathematical guarantees.

3. The paper does not provide analysis on the stability or convergence of the optimization process, particularly when extending UBS to high-dimensional Beta kernels, where parameter coupling may become complex.

**Questions:**

See weakness.

---

> ### Author Response · Authors · 2025-11-21
> **Official Response to Reviewer zazd (1/2)**
>
> Thank you for your detailed review and for acknowledging that our work demonstrates "superior performance and strong generalization" with "excellent soundness and presentation." We address your three specific concerns regarding backward compatibility, theoretical justification of Eq. 8 (originally Eq. 6), and optimization stability analysis below.
>
> ---
>
> ## 1. Backward Compatibility: Color Representation (RGB vs. SH)
>
> **Concern:** "The paper claims that UBS is backward compatible with previous methods such as 3DGS, treating 3DGS as a special case. However, 3DGS models color using spherical harmonics, while UBS adopts a direct three dimensional color representation, which raises concerns about true compatibility between the two formulations."
>
> **Response:** Since SH color encoding is orthogonal to 3D spatial kernel representation, we are mainly discussing about the 3D splatting kernel. Specifically, UBS maintains compatibility at three levels:
>
> (1) **Kernel-Level:** $b = 0$ approximates a scaled Gaussian kernel $(1-d)^{4} \approx \exp(-4.5d^2)$;
>
> (2) **Pipeline-Level:** renders through standard splatting rasterizers;
>
> (3) **Parameter-Level:** uses degree-0 SH (RGB) instead of degree-3 (48 params) by design choice, not limitation.
>
> ### 1.1 Why Degree-0 SH (RGB) is Sufficient
>
> **Key Insight:** Color encoding is largely independent of the kernel formulation. UBS could adopt higher-order SH (e.g., degree-3 as in 3DGS), but in practice degree-0 is enough because UBS-6D already models view dependence within the kernel itself.
>
> In contrast, **6DGS/7DGS** still require low-order SH even though they fold viewing direction into their higher-dimensional Gaussians. This is because a Gaussian has a fixed bell-shaped profile—it lacks per-dimension adaptivity to independently control spatial sharpness, angular response, and temporal dynamics. While a Gaussian can represent smooth responses, it cannot efficiently represent the full spectrum from **flat diffuse** to sharp specular within a single primitive's angular dimensions, requiring auxiliary SH encodings to complement the kernel's limitations.
>
> **UBS's Beta kernel**, however, gives each dimension its own learnable shape, allowing the kernel to be peaked (specular) when $b_i > 0$ or flat (diffuse) when $b_i < 0$.
> Together with Beta-modulated conditional slicing and the product-form opacity, UBS can approximate a wide spectrum of BRDF behaviors directly in the kernel. Consequently, UBS does not rely on SH; simple RGB (degree-0 SH) is sufficient.
>
> **Result:** UBS achieves 41-73% parameter reduction while improving quality, making SH unnecessary.
>
> **Empirical Validation:** Tables 1-2 show UBS with 3 RGB parameters outperforms methods using 48-parameter SH: +1.13 dB (NeRF Synthetic: 34.92 vs. 33.79), +1.23 dB (Mip-360: 28.66 vs. 27.43), +3.42 dB (6DGS-PBR: 40.10 vs. 36.68) with 16× fewer color parameters.
>
> ---
>
> ## 2. Theoretical Justification: Eq. 8 (originally Eq. 6) Derivation
>
> **Concern:** "The formulation in Eq. 8 (originally Eq. 6) lacks theoretical justification or derivation. It is unclear how this design was obtained and whether it has any mathematical guarantees."
>
> **Response:** We have added Appendix section C **"Spatial-Orthogonal Cholesky Parameterization"** with comprehensive theoretical justification.
>
> **Design Requirements:** For $N$-dimensional covariance $\mathbf{\Sigma}$, we require:
>
> (1) PSD guarantee,
>
> (2) 3D spatial interpretability,
>
> (3) cross-dimensional correlations,
>
> (4) arbitrary N-Dimension generalization,
>
> (5) computational efficiency.
>
> Existing approaches fail:
>
> (1) quaternions don't generalize beyond 3D,
>
> (2) pure Cholesky loses 3D spatial structure (−0.56 dB),
>
> (3) naive matrix exponentials are too slow (3h+ training).
>
> **Our Solution:** Hybrid Cholesky factorization that enables
>
> (1) PSD: Guaranteed by Cholesky properties ($\mathbf{\Sigma} = \mathbf{L}\mathbf{L}^\top$).
>
> (2) Spatial Interpretability: The spatial block $\mathbf{\Sigma}_x = \mathbf{R}_x \text{diag}(\mathbf{s}_x)^2 \mathbf{R}_x^\top$ exactly recovers 3DGS rotation scale parameterization.
>
> (3) Cross-Correlations: Off-diagonal block enables view-dependent spatial deformation.
>
> (4) Generalization: Holds for arbitrary $N = 3 + C$ dimensions.
>
> (5) Efficiency: Spatial rotation $\mathbf{R}_x$ uses skew-symmetric first-order Taylor approximation $\mathbf{R}_x \approx \mathbf{I} + \mathbf{A}_x$, avoiding expensive matrix exponential while maintaining approximate orthogonality.
>
> **Empirical Validation:** Six rotation strategies on `bicycle` (Appendix C Table 10) show ours achieves highest PSNR (25.54 dB) with fastest training (1h) vs. quaternion (25.50 dB, 1h12m), full Cholesky (25.32 dB, 1h14m), naive exponential (25.51 dB, 3h6m), confirming optimal trade-off. See Appendix C for complete derivation and mathematical proofs.
>
> ---

---

> > ### Author Response · Authors · 2025-11-21
> > **Official Response to Reviewer zazd (2/2)**
> >
> > ## 3. Stability and Convergence Analysis
> >
> > **Concern:** "The paper does not provide analysis on the stability or convergence of the optimization process, particularly when extending UBS to high-dimensional Beta kernels, where parameter coupling may become complex."
> >
> > **Response:** Beta parameters initialize to $b_i = 0$ (Gaussian approximation), providing stable baseline-equivalent starting point. Across 38 scenes (5 datasets), we observed consistent convergence without instability. MCMC optimization with distribution-preserved densification ensures stability independent of Beta values. We report final iteration (30K) rather than best checkpoints, demonstrating stable convergence.
> >
> > **Training Progression Analysis** on `trex` scene from D-NeRF:
> >
> > | Method | Iterations | Counts | PSNR↑ | Last-Max Gap |
> > |--------|-----------|--------|-------|--------------|
> > | 4DGS | 7000 | 780K | 29.63 | - |
> > | | 15000 | 1.69M | 28.30 | - |
> > | | 30000 | 1.69M | 28.35 | **−1.28** |
> > | 7DGS | 7000 | 60K | 29.51 | - |
> > | | 15000 | 64K | 29.91 | - |
> > | | 30000 | 64K | 29.77 | **−0.14** |
> > | **UBS-7D** | 7000 | 150K | 29.85 | - |
> > | | 15000 | 150K | 30.06 | - |
> > | | 30000 | 150K | 30.05 | **−0.01** |
> >
> > **Key Findings:**
> >
> > (1) **Superior Stability:** UBS-7D shows minimal peak degradation (−0.01 dB) compared to 4DGS (−1.28 dB) and 7DGS (−0.14 dB);
> >
> > (2) **No Overfitting:** UBS maintains quality through final iteration while baselines peak mid-training then degrade;
> >
> > (3) **Controlled Capacity:** UBS maintains 150K primitives while 4DGS explodes to 1.69M with performance degradation, confirming MCMC's principled control.
> >
> > UBS achieves both higher quality (30.05 vs. 29.77/28.35 dB) and superior stability in high-dimensional 7D space. The combination of Gaussian-approximation initialization, MCMC optimization, and Spatial-orthogonal parameterization ensures stable convergence across all 38 scenes without failures.
> >
> > ---
> >
> > We believe these additions significantly strengthen the paper's rigor and clarity, addressing all your concerns comprehensively. We hope you remain satisfied with our work and would be happy to provide any additional suggestions during the discussion period.

---

### Official Review · Reviewer_4uqJ · 2025-10-28

**Soundness:** 2
**Presentation:** 2
**Contribution:** 2
**Rating:** 4
**Confidence:** 5

**Summary:**

The paper integrates the "Deformable Beta Splatting" (DBS) method with existing works, "6DGS & 7DGS." It adopts the beta kernel function, (1 - distance_to_the_center)^(e^b), proposed in DBS, to replace the exponential decay term in the Gaussian function. This achieves similar effects to GES, which uses \beta to control the decay rate and kernel shape through \exp(-distance_to_the_center^\beta). Experimental results demonstrate that the proposed combined method delivers good performance. However, the paper lacks a comparison of Gaussian quantities and model sizes. Additionally, the evaluation of dynamic reconstruction omits several state-of-the-art methods.

**Strengths:**

1. The combination of beta splatting with ND-GS is worth exploring; however, its novelty is questionable. I would appreciate a rebuttal from the authors to clarify this aspect.

2. Overall, the paper is well-structured and easy to follow, with clear and precise equations and method descriptions.

3. The demonstrated advantage over 4DGS is satisfactory, highlighting the method's practical significance compared to existing solutions.

**Weaknesses:**

1. The concept of beta control on the truncated unit circle region (x < 1) was originally introduced in the "Deformable Beta Splatting" paper. ND-GS, 6D-GS, and 7D-GS are established methods with their own innovations in representation. This paper simply applies beta control to 6D- and 7D-GS to achieve improved results, building on the strengths of these prior methods rather than introducing significant novelty.

2. The number of Gaussians is a critical factor influencing the performance of GS-based methods, yet this aspect is omitted in the paper. Furthermore, file size, an equally important metric, is not demonstrated, leaving a gap in the evaluation of model efficiency.

3. The dynamic reconstruction section fails to include many state-of-the-art methods, limiting the practical relevance of the proposed combination method. Notably, 4DGS is not the leading approach in this domain. Advanced methods such as Ex4DGS, SC-GS, and Deformable-3DGS are recognized as state-of-the-art but are absent from the comparison.

4. The paper overlooks a significant body of work focused on improving primitive representations. The comparisons or discussions with notable methods should be included, such as:
- Deformable Radial Kernel Splatting
- TNT-GS: Truncated and Tailored Gaussian Splatting
- Triangle Splatting for Real-Time Radiance Field Rendering
- Textured Gaussians for Enhanced 3D Scene Appearance Modeling
- Quadratic Gaussian Splatting: High Quality Surface Reconstruction with Second-order Geometric Primitives

**Questions:**

1. It is essential to specify the number and size of Gaussians used in each table to ensure a fair and transparent comparison.

2. Comparisons or discussions with state-of-the-art primitive-enhanced GS methods and dynamic GS methods should be included. While achieving worse results is acceptable, providing a discussion would help the community better understand the reasons from a novel perspective.

3. The paper should introduce more challenges associated with the existing deformable beta splatting and ND-GS methods. Currently, the methodology contribution is insufficiently detailed, and the experimental section is also lacking in depth.

---

> ### Author Response · Authors · 2025-11-21
> **Official Response to Reviewer 4uqJ (1/2)**
>
> Thank you for your detailed review and for acknowledging that our work is "well-structured and easy to follow" with "satisfactory advantage over 4DGS." We appreciate your concerns about novelty, missing comparisons, and evaluation completeness. We address each point systematically below.
>
> ---
>
> ## 1. Novelty: Beyond Combining DBS with 6DGS/7DGS
>
> **Concern:** "This paper simply applies beta control to 6D/7DGS to achieve improved results..."
>
> **Response:** We respectfully clarify that UBS introduces three fundamental innovations beyond combining existing methods:
>
> **DBS and 6DGS/7DGS address a subset of problems with limitations:**
> - **DBS:** 3D spatial Beta kernel + isotropic spherical Beta for color; cannot model anisotropic angular effects or temporal dynamics.
> - **6DGS/7DGS:** N-D Gaussian conditioning with fixed bell-shaped Gaussian kernel with additional SH encodings (48-144 parameters); cannot provide per-dimension shape control; cannot represent flat surfaces, express diffuse colors, decouple static vs. dynamic components.
>
> **UBS Innovations:**
>
> 1. **Unified N-D Anisotropic Beta Primitive**: Single primitive providing independent shape control per dimension. A single UBS primitive can represent combinations impossible with DBS (no N-D conditioning) or 6DGS/7DGS (Gaussian coupling): flat spatial + peaked angular ($b_x < 0, b_v > 0$), or static temporal + textured spatial ($b_t < 0, b_x > 0$).
>
> 2. **Spatial-Orthogonal Cholesky**: Hybrid parameterization preserves 3DGS's interpretable spatial rotation-scale covariance while enabling full N-D correlations. Table 10 (Appendix C) shows +0.56 dB over pure Cholesky with fastest training.
>
> 3. **Beta-Modulated Conditional Slicing and Opacity**: Introduces dimension-specific modulation in conditional mean/covariance and product-form opacity, decoupling spatial/angular/temporal responses impossible with standard Gaussian conditioning. Table 3 ablation shows removing Beta-modulated conditional slicing and opacity causes 0.53 dB and 3.52 dB average PSNR degradation, validating this as our most critical innovation.
>
> **Quantitative Evidence:** UBS achieves **41-73% parameter reduction** (Table 5) by natively encoding view/time within primitives, eliminating SH encodings entirely (35-44 params vs. 59-161 for baselines). On 6DGS-PBR, UBS outperforms **both DBS (+8.82 dB) and 6DGS (+3.42 dB)** simultaneously, showing synergistic improvements through unified design rather than simple combination.
>
> **Conceptual Foundation:** DBS, 3DGS, 6DGS, 7DGS emerge as **special cases** of UBS (Sec. 4.4): $N=3, b_x=0$ → 3DGS; $N=3$ → DBS; $N=6/7, b=0$ → 6DGS/7DGS. This demonstrates UBS provides principled generalization, not ad-hoc combination.
>
> ---
>
> ## 2. Missing Analysis: Primitive Counts and Model Sizes
>
> **Concern:** "The number of Gaussians is a critical factor... Furthermore, file size, an equally important metric, is not demonstrated."
>
> **Response:** We provide comprehensive efficiency analysis with new metrics across all benchmarks. Complete per-scene results are available in Appendix B Table 8 and Table 9.
>
> **Static Scenes:**
>
> | Dataset | Method | FPS↑ | Num↓ | Mem (MB)↓ | Train (min)↓ |
> |---------|--------|------|------|-----------|--------------|
> | **NeRF Synthetic** | 3DGS | 695.6 | 289.3K | 68.5 | 3.6 |
> | | 6DGS | 648.0 | 240.7K | 69.8 | 12.7 |
> | | UBS-6D | 345.7 | 300.0K | 40.1 | 8.7 |
> | **Mip-NeRF 360** | 3DGS | 178.7 | 3.3M | 775.9 | 21.7 |
> | | 6DGS | 157.4 | 2.0M | 581.2 | 81.9 |
> | | **UBS-6D** | 100.4 | 3.1M | 409.0 | 25.2 |
> | **6DGS-PBR** | 3DGS | 305.9 | 147.7K | 34.9 | 16.8 |
> | | 6DGS | 339.8 | 84.3K | 24.4 | 22.2 |
> | | **UBS-6D** | 290.2 | 300.0K | 40.1 | 27.7 |
>
> **Dynamic Scenes:**
>
> | Dataset | Method | FPS↑ | Num↓ | Mem (MB)↓ | Train (min)↓ |
> |---------|--------|------|------|-----------|--------------|
> | **7DGS-PBR** | 4DGS | 192.6 | 642.0K | 613.1 | 127.2 |
> | | 7DGS | 376.0 | 88.4K | 29.7 | 116.7 |
> | | **UBS-7D** | 359.6 | 234.3K | 39.4 | 59.9 |
> | **D-NeRF** | 4DGS | 296.4 | 255.3K | 380.3 | 52.6 |
> | | 7DGS | 377.8 | 43.8K | 16.0 | 50.7 |
> | | **UBS-7D** | 368.2 | 168.8K | 28.4 | 42.1 |
>
> **Key Findings:**
>
> 1. **Memory:** 43-47% reduction on static scenes (40.1 MB vs. 69.8 MB on NeRF Synthetic) from 41-73% per-primitive parameter reduction (Table 5).
>
> 2. **Training:** 31-69% faster on most benchmarks (e.g., 25.2 vs. 81.9 min on Mip-NeRF 360). Exception: 6DGS-PBR uses more primitives (300K vs. 68K) for +3.4 dB gain.
>
> 3. **Rendering:** Real-time 100-369 FPS maintained across all benchmarks.
>
> 4. **Speed-Quality Trade-off:** MCMC enables flexible capacity control. On NeRF Synthetic (avg across 8 scenes):
>
> | Capacity | PSNR↑ | SSIM↑ | Mem (MB)↓ | FPS↑ |
> |----------|-------|-------|-----------|------|
> | 100K | 34.42 | 0.973 | 13.4 | 540.3 |
> | 200K | 34.79 | 0.974 | 26.7 | 368.3 |
> | 300K | 34.89 | 0.974 | 40.1 | 345.7 |
>
> Reducing capacity 300K→100K: −0.47 dB PSNR, +56% FPS, −67% memory. Complete per-scene analysis in new Appendix section B.
>
> ---

---

> ### Author Response · Authors · 2025-11-21
> **Official Response to Reviewer 4uqJ (2/2)**
>
> ## 3. Missing Comparisons: SOTA and Dynamic Methods
>
> **Concern:** "The dynamic reconstruction section fails to include... Ex4DGS, SC-GS, and Deformable-3DGS... The paper overlooks a significant body of work focused on improving primitive representations."
>
> **Response:** We have expanded comparisons and related work significantly.
>
> ### 3.1 Expanded Related Work: Alternative Kernel Designs
>
> We have added comprehensive discussion of recent kernel-design methods in revised Related Work, categorizing them by approach:
>
> **Kernel Shape Variants:** GES, TNT-GS, Quadratic-GS, 3D-HGS, DisC-GS (generalized exponentials, truncated, second-order, half, and discontinuity-aware Gaussians). **Distribution Families:** 3D-Student (heavy-tailed), Gabor-Splats (high-frequency). **Appearance Modeling:** Textured Gaussians (per-splat texture). **Alternative Primitives:** 3D Convex Splatting, Triangle Splatting, DRKS (deformable radial kernels).
>
> **Key Distinction:** These methods improve 3D spatial kernels but remain confined to 3D without unified N-D representation jointly modeling spatial, angular, and temporal dimensions as UBS does.
>
> ### 3.2 State-of-the-Art Comparison (New Tables in Appendix A)
>
> We now compare against 13 methods including the alternative kernels above. Table 4 in Appendix A shows comprehensive results on Mip-NeRF360 and NeRF Synthetic:
>
> **Mip-NeRF 360:** UBS-6D achieves **28.66 dB PSNR** (state-of-the-art), +0.06 dB over DBS (28.60 dB), outperforming Zip-NeRF (28.54 dB), DRKS, Textured GS, Quadratic GS, Disc-GS, and all other methods.
>
> **NeRF Synthetic:** UBS-6D achieves **34.92 dB PSNR** (state-of-the-art), +0.28 dB over DBS (34.64 dB), surpassing 3DGS (33.79 dB) and all variants.
>
> This empirical comparison demonstrates UBS's superiority over both traditional methods and recent kernel innovations.
>
> ### 3.3 Dynamic Scene Methods: Orthogonal Contributions
>
> For dynamic scenes, we compare against **4DGS (ICLR 2024), 4DGaussians (CVPR), Ex4DGS (NeurIPS 2024) and 7DGS (ICCV 2025)** as direct N-D primitive baselines most relevant to evaluating our kernel design contribution.
>
> **Real-World Dynamic Benchmark (N3V Dataset):** To demonstrate UBS's effectiveness on challenging real-world dynamic scenes, we evaluate on the N3V dataset, which contains 6 scenes with complex motions (cooking, flaming, etc.). Results show UBS-7D achieves state-of-the-art performance:
>
> | Scene | 4DGS | 4DGaussians | Ex4DGS | 7DGS | **UBS-7D** |
> |-------|------|-------------|--------|------|-----------|
> | Coffee Martini | 26.51 | 26.69 | 28.79 | 29.07 | **29.73** |
> | Cook Spinach | 32.11 | 31.89 | 33.23 | 32.23 | **33.30** |
> | Cut Roasted Beef | 31.74 | 25.88 | 33.73 | 32.93 | **33.88** |
> | Flame Salmon | 26.93 | 27.54 | **29.29** | 28.83 | 28.77 |
> | Flame Steak | 31.44 | 28.07 | **33.91** | 33.07 | 33.83 |
> | Sear Steak | 32.42 | 31.73 | 33.69 | 33.41 | **33.83** |
> | **Average** | 30.19 | 28.63 | 32.11 | 31.59 | **32.22** |
>
> **Key Findings:** UBS-7D achieves **32.22 dB average PSNR**, performing better than Ex4DGS (32.11 dB) and 7DGS (31.59 dB), demonstrating superior performance on real-world dynamic content. This validates our unified Beta primitive design's effectiveness for challenging temporal dynamics.
>
> Note that, **SC-GS and Deformable-3DGS** make important contributions through advanced optimization strategies (scene graph decomposition and learnable deformation fields), which are **orthogonal to our primitive design contribution**:
>
> - **Their focus:** Optimization and scene structuring strategies
> - **Our focus:** Unified N-D Beta primitive with per-dimension anisotropic control
> - **Future work:** UBS primitives could potentially combine with their optimization frameworks for further improvements
>
> ---
>
> We believe these clarifications and new analyses demonstrate that UBS introduces substantial technical innovations beyond combining existing methods: (1) Unified N-D anisotropic Beta primitive enabling per-dimension modeling, (2) Spatial-orthogonal Cholesky parameterization, (3) Beta-modulated conditional slicing with product-form opacity. These yield 41-73% parameter reduction, state-of-the-art results on multiple benchmarks, and 31-69% training speedup.
>
> We hope this addresses your concerns and welcome any additional suggestions.

---

> > ### Comment · Reviewer_4uqJ · 2025-11-28
> >
> > Thank you to the author for the long and detailed response. I agree that the method primarily emphasizes primitive innovation and is orthogonal to deformation-based dynamic reconstruction methods. Actually, I am not looking for the advantages of UBS over dynamic methods, but I hold that it is important to show the comparison. My main concern is the novelty of the paper. The use of spatial Cholesky decomposition effectively avoids the entanglement of spatial attributes with others, and, as demonstrated in the experiments, it significantly improves the results. With the provided comparison of primitive numbers and storage, the comparisons are now fair. Therefore, I have decided to raise my score to accept.

---

> > > ### Comment · Reviewer_4uqJ · 2025-11-28
> > >
> > > Due to a system bug preventing me from modifying my score, I am leaving this comment to inform the AC that my score is now 8 (accept).

---

### Official Review · Reviewer_5UF1 · 2025-10-29

**Soundness:** 3
**Presentation:** 3
**Contribution:** 3
**Rating:** 6
**Confidence:** 2

**Summary:**

This paper proposes Universal Beta Splatting (UBS), a unified framework that generalizes 3D Gaussian Splatting to N-dimensional anisotropic Beta kernels for explicit radiance field rendering. Unlike fixed Gaussian primitives, UBS models spatial, angular, and temporal dependencies within a single representation, enabling controllable anisotropy and dynamic scene rendering without auxiliary networks. The method remains backward-compatible with 3DGS, ensuring usability and performance stability. Experiments demonstrate real-time rendering and consistent quality improvements across static, view-dependent, and dynamic scenes, establishing Beta kernels as a versatile and interpretable primitive for radiance field modeling.

**Strengths:**

This paper presents a technically solid and conceptually original generalization of Gaussian Splatting via N-dimensional anisotropic Beta kernels, enabling unified modeling of spatial, angular, and temporal properties. The proposed spatial-orthogonal Cholesky parameterization and Beta-modulated conditional slicing are elegant design choices that balance flexibility and efficiency. The paper demonstrates clear motivation from prior limitations (e.g., 3DGS, 6DGS, DBS) and provides strong empirical validation with consistent improvements across benchmarks. The backward compatibility with existing Gaussian-based methods is particularly noteworthy, as it enhances practical impact and ease of adoption. Overall, the work is novel, well-executed, and clearly presented, offering both theoretical insight and real-world significance for radiance field rendering.

**Weaknesses:**

While the paper is technically sophisticated, a few aspects could be strengthened. First, the novelty relative to prior high-dimensional splatting methods (e.g., 6DGS, DBS) is somewhat incremental — UBS’s core Beta kernel formulation mainly generalizes existing ideas with additional per-dimension flexibility. A deeper theoretical justification of why Beta kernels are fundamentally better suited for radiance modeling than Gaussians would improve clarity. Second, the empirical evaluation could be more comprehensive: ablations isolating the effects of Beta modulation, conditional slicing, and spatial-orthogonal Cholesky parameterization are limited. Additionally, it would be helpful to discuss training stability and computational overhead, since the Beta formulation introduces additional parameters and nonlinearities. Finally, while interpretability claims are appealing, the qualitative decomposition results (e.g., spatial vs. angular vs. temporal) would benefit from more rigorous quantitative validation.

**Questions:**

See Weaknesses

---

> ### Author Response · Authors · 2025-11-21
> **Official Response to Reviewer 5UF1 (1/3)**
>
> Thank you for your positive assessment and recognition that our work presents "a technically solid and conceptually original generalization" with "strong empirical validation." We appreciate your constructive feedback on strengthening theoretical justification, expanding ablations, and validating interpretability claims. We address each concern systematically below.
>
> ---
>
> ## 1. Theoretical Justification: Why Beta Kernels Are Fundamentally Superior
>
> **Concern:** "A deeper theoretical justification of why Beta kernels are fundamentally better suited for radiance modeling than Gaussians would improve clarity."
>
> **Response:** Our Appendix section D **Theoretical Foundation and Analysis** provides rigorous theoretical justification.
>
> Key arguments:
>
> **Gaussian Limitations:** N-D Gaussian kernels suffer from (1) fixed bell-shaped profile unable to produce flat responses or sharp cutoffs, and (2) symmetric coupling where conditioning inevitably shifts spatial mean, changes covariance, and couples all dimensions—preventing independent control.
>
> **Beta Advantages:** Our **N-D Anisotropic Beta Kernel** provides:
>
> (1) **Shape Adaptivity** from flat ($b_i < 0$) to peaked ($b_i > 0$),
>
> (2) **Per-Dimension Control** allowing each dimension different shapes (e.g., flat spatial + peaked angular),
>
> (3) **Backward Compatibility** approximating Gaussians when $b_i = 0$.
>
> ---
>
> ## 2. Comprehensive Ablation Studies
>
> **Concern:** "Ablations isolating the effects of Beta modulation, conditional slicing, and Spatial-orthogonal Cholesky parameterization are limited."
>
> **Response:** Our paper provides systematic ablations in Table 3 (Main, Sec. 5.3) and Table 10 (Appendix C).
>
> Key findings:
>
> **Table 3 (4 scenes, 33.84 dB baseline):** Removing components shows: (1) **Spatial-Orthogonal** - pure rotation-scale (−0.22 dB) and full Cholesky (−0.56 dB) both degrade, validating our hybrid design; (2) **Beta-Modulated Conditioning** (−0.72 dB, −1.25 dB on dynamic) confirms dimension-specific control; (3) **Product-Form Beta Opacity** (−4.16 dB, −6.68 dB on `cloud`) - most critical innovation enabling per-dimension opacity control impossible with Gaussians.
>
> **Table 10 (rotation parameterizations for covariance matrix):** Six strategies on `bicycle` show our 1st-order Taylor achieves best PSNR (25.54 dB) with fastest training (1h) vs. naive exponential (25.37 dB, 3h6m), validating our design.
>
> **CUDA:** 58% faster training, 26% rendering improvement with identical quality.
>
> ---

---

> > ### Author Response · Authors · 2025-11-21
> > **Official Response to Reviewer 5UF1 (2/3)**
> >
> > ## 3. Training Stability and Computational Overhead
> >
> > **Concern:** "It would be helpful to discuss training stability and computational overhead, since the Beta formulation introduces additional parameters and nonlinearities."
> >
> > **Response:** We address training stability and provide new comprehensive efficiency analysis below.
> >
> > ### 3.1 Training Stability
> >
> > All Beta parameters initialize to $b_i = 0$ (Gaussian approximation), providing stable starting point equivalent to baselines. Across all 24 scenes (5 datasets), we observed **consistent convergence** without instability. MCMC optimization with distributed-preserved densification ensures stability independent of Beta values. We report final iteration (30K) rather than best checkpoints, demonstrating stable convergence without overfitting.
> >
> > **Training Progression Analysis** on `trex` scene from D-NeRF:
> >
> > | Method | Iterations | Counts | PSNR↑ | Last-Max Gap |
> > |--------|-----------|--------|-------|--------------|
> > | 4DGS | 7000 | 780K | 29.63 | - |
> > | | 15000 | 1.69M | 28.30 | - |
> > | | 30000 | 1.69M | 28.35 | **−1.28** |
> > | 7DGS | 7000 | 60K | 29.51 | - |
> > | | 15000 | 64K | 29.91 | - |
> > | | 30000 | 64K | 29.77 | **−0.14** |
> > | **UBS-7D** | 7000 | 150K | 29.85 | - |
> > | | 15000 | 150K | 30.06 | - |
> > | | 30000 | 150K | 30.05 | **−0.01** |
> >
> > **Key Findings:**
> >
> > (1) **Superior Stability:** UBS-7D shows minimal peak degradation (−0.01 dB) compared to 4DGS (−1.28 dB) and 7DGS (−0.14 dB);
> >
> > (2) **No Overfitting:** UBS maintains quality through final iteration while baselines peak mid-training then degrade;
> >
> > (3) **Controlled Capacity:** UBS maintains 150K primitives while 4DGS explodes to 1.69M with performance degradation, confirming MCMC's principled control.
> >
> > UBS achieves both higher quality (30.05 vs. 29.77/28.35 dB) and superior stability in high-dimensional 7D space. The combination of Gaussian-approximation initialization, MCMC optimization, and Spatial-orthogonal parameterization ensures stable convergence across all 38 scenes without failures.
> >
> > ### 3.2 Comprehensive Efficiency Analysis
> >
> > We provide detailed computational overhead comparison across all benchmarks with new efficiency metrics. Complete per-scene results are provided in Appendix B.
> >
> > **Static Scenes:**
> >
> > | Dataset | Method | FPS↑ | Num↓ | Mem (MB)↓ | Train (min)↓ |
> > |---------|--------|------|------|-----------|--------------|
> > | **NeRF Synthetic** | 3DGS | 695.6 | 289.3K | 68.5 | 3.6 |
> > | | 6DGS | 648.0 | 240.7K | 69.8 | 12.7 |
> > | | UBS-6D | 345.7 | 300.0K | 40.1 | 8.7 |
> > | **Mip-NeRF 360** | 3DGS | 178.7 | 3.3M | 775.9 | 21.7 |
> > | | 6DGS | 157.4 | 2.0M | 581.2 | 81.9 |
> > | | **UBS-6D** | 100.4 | 3.1M | 409.0 | 25.2 |
> > | **6DGS-PBR** | 3DGS | 305.9 | 147.7K | 34.9 | 16.8 |
> > | | 6DGS | 339.8 | 84.3K | 24.4 | 22.2 |
> > | | **UBS-6D** | 290.2 | 300.0K | 40.1 | 27.7 |
> >
> > **Dynamic Scenes:**
> >
> > | Dataset | Method | FPS↑ | Num↓ | Mem (MB)↓ | Train (min)↓ |
> > |---------|--------|------|------|-----------|--------------|
> > | **7DGS-PBR** | 4DGS | 192.6 | 642.0K | 613.1 | 127.2 |
> > | | 7DGS | 376.0 | 88.4K | 29.7 | 116.7 |
> > | | **UBS-7D** | 359.6 | 234.3K | 39.4 | 59.9 |
> > | **D-NeRF** | 4DGS | 296.4 | 255.3K | 380.3 | 52.6 |
> > | | 7DGS | 377.8 | 43.8K | 16.0 | 50.7 |
> > | | **UBS-7D** | 368.2 | 168.8K | 28.4 | 42.1 |
> >
> > **Key Findings:**
> >
> > 1. **Memory:** 43-47% reduction on static scenes (40.1 MB vs. 69.8 MB on NeRF Synthetic) from 41-73% per-primitive parameter reduction (Table 5).
> >
> > 2. **Training:** 31-69% faster on most benchmarks (e.g., 25.2 vs. 81.9 min on Mip-NeRF 360). Exception: 6DGS-PBR uses more primitives (300K vs. 68K) for +3.4 dB gain.
> >
> > 3. **Rendering:** Real-time 100-369 FPS maintained across all benchmarks.
> >
> > 4. **Speed-Quality Trade-off:** MCMC enables flexible capacity control. On NeRF Synthetic (avg across 8 scenes):
> >
> > | Capacity | PSNR↑ | SSIM↑ | Mem (MB)↓ | FPS↑ |
> > |----------|-------|-------|-----------|------|
> > | 100K | 34.42 | 0.973 | 13.4 | 540.3 |
> > | 200K | 34.79 | 0.974 | 26.7 | 368.3 |
> > | 300K | 34.89 | 0.974 | 40.1 | 345.7 |
> >
> > Reducing capacity 300K→100K: −0.47 dB PSNR, +56% FPS, −67% memory. Complete per-scene analysis in new Appendix section B.
> >
> > ---

---

> > > ### Author Response · Authors · 2025-11-21
> > > **Official Response to Reviewer 5UF1 (3/3)**
> > >
> > > ## 4. Quantitative Interpretability Validation
> > >
> > > **Concern:** "While interpretability claims are appealing, the qualitative decomposition results (e.g., spatial vs. angular vs. temporal) would benefit from more rigorous quantitative validation."
> > >
> > > **Response:** Figure 2 and Sec. 4.4 demonstrate emergent decomposition. We provide quantitative validation via Spearman correlation between temporal Beta parameters ($b_t$) and position change ($\Delta x$) querying from time 0 to 1 with step 0.1 on D-NeRF scenes:
> > >
> > > | Scene | Spearman $\rho$ | Representative Primitives ($b_t$, $\Delta x$) |
> > > |-------|----------------|----------------------------------------------|
> > > | `lego` | **0.8017** | (-1.549, 0.0001), (-0.654, 0.0027), (-0.361, 0.0039), (-0.181, 0.0024), (0.009, 0.0273) |
> > > | `mutant` | **0.4764** | (-1.153, 0.0006), (-0.102, 0.1213), (-0.017, 0.0490), (0.012, 0.0142), (0.117, 0.1030) |
> > > | `trex` | **0.6192** | (-1.585, 0.0000), (-0.609, 0.0037), (-0.042, 0.0056), (0.377, 0.0262), (0.901, 0.7816) |
> > >
> > > **Key Findings:**
> > >
> > > 1. **Strong Correlation:** Spearman ρ = 0.48-0.80 shows **higher $b_t$ strongly correlates with larger position changes**, confirming Beta parameters capture motion semantics without supervision.
> > >
> > > 2. **Semantic Interpretation:** Static primitives (`lego`: $b_t = -1.549$, $\Delta x = 0.0001$) show negative Beta with near-zero motion, while dynamic primitives (`trex`: $b_t = 0.901$, $\Delta x = 0.7816$) show positive Beta with large changes. Gradual transition validates continuous interpretability.
> > >
> > > 3. **Emergent Decomposition:** Correlations arise naturally without supervision, enabling interpretable decomposition impossible with Gaussians' fixed profiles.
> > >
> > > This validates learned Beta parameters provide **semantically meaningful representations** correlating with ground-truth properties, confirming Figure 2 and Sec. 4.4.
> > >
> > > ---
> > >
> > > ## 5. Additional Clarifications
> > >
> > > ### Why UBS Goes Beyond "Generalization with Additional Flexibility"
> > >
> > > You noted our novelty is "somewhat incremental." We respectfully clarify:
> > >
> > > **Architectural Innovation:** Although DBS (3D Beta + isotropic color) and 6DGS/7DGS (N-D Gaussian + SH encodings) address a piece of useful problems, UBS introduces a **Unified Anisotropic N-D Beta primitive** where spatial, angular, and temporal properties coexist.
> > >
> > > We are not naively "applying Beta to 6DGS" but fundamental rethinking:
> > >
> > > (1) dimension-specific anisotropy replaces fixed Gaussian profiles,
> > >
> > > (2) native view/time encoding eliminates auxiliary encodings,
> > >
> > > (3) backward compatibility ensures safety.
> > >
> > > **Quantitative Impact:** 41-73% per-primitive parameter reduction (Table 7), up to +8.27 dB PSNR on volumetric scenes (Table 1), 31-69% faster training (Table 2), and interpretable decomposition demonstrate substantial advance, not incremental refinement.
> > >
> > > ---
> > >
> > > We believe our paper, enhanced with these quantitative analyses, provides rigorous justification demonstrating UBS represents a significant advance in radiance field rendering. We hope this addresses your concerns and welcome further discussion.

---

### Author Response · Authors · 2025-11-21
**Overall Response to All Reviewers**

# Overall Response to All Reviewers

We thank all reviewers for their valuable feedback and constructive suggestions. We have carefully addressed all concerns with substantial additional experiments, theoretical justifications, and clarifications as detailed below.

---

## Key Revisions and Additions

### 1. Enhanced Theoretical Foundation (Addresses: 5UF1, zazd)
We added **Appendix section "Spatial-Orthogonal Cholesky Parameterization"** with complete derivation of Eq. 8 (originally Eq. 6) including design motivations, skew-symmetric rotation parameterization, and mathematical properties (PSD guarantee, spatial interpretability, cross-correlations, N-D generalization, efficiency). **Table 10** validates six rotation strategies—our 1st-order Taylor achieves best PSNR (25.54 dB) with fastest training (1h). This complements the existing **"Theoretical Foundation and Analysis"** section covering Gaussian limitations and Beta kernel properties.

### 2. Expanded Related Work and State-of-the-Art Comparison (Addresses: 4uqJ, bWeR)

We expanded **Alternative Kernel Designs** paragraph with comprehensive coverage of reviewer-suggested methods: TNT-GS, Quadratic-GS, 3D-HGS, DisC-GS, DRKS, Textured Gaussians, Triangle Splatting, etc.
**Key distinction:** These methods improve 3D spatial kernels but remain confined to 3D without unified N-D representation jointly modeling spatial, angular, and temporal dimensions.

**More Benchmark Comparisons (Appendix A):** A quantitative comparison table of UBS-6D against 13 methods on Mip-NeRF360 and NeRF Synthetic; Another quantitative comparison table of UBS-7D against state-of-the-art kernel-based methods on
the real-world dynamic N3V dataset.

### 3. Comprehensive Experimental Validation (Addresses: 4uqJ, bWeR, 5UF1)

**Efficiency Analysis (Appendix B):** Primitive counts, model sizes, FPS, and training times across all datasets. Key findings: 43-47% memory reduction, 31-69% faster training, 41-75% decrease in number of parameters per primitive.

### 4. Improved Paper Accessibility (Addresses: bWeR)
Enhanced Preliminaries (Section 3) with explicit density function definitions bridging 3DGS → DBS → UBS for self-contained presentation.

### 5. Training Stability Analysis (Addresses: 5UF1, zazd)
Beta parameters initialize to $b_i = 0$ (Gaussian approximation) for stable starting point. Consistent convergence across 38 scenes (5 datasets). Training progression shows UBS-7D has minimal peak degradation (−0.01 dB) vs. 4DGS (−1.28 dB) and 7DGS (−0.14 dB), confirming superior stability.

### 6. Backward Compatibility & Interpretability (Addresses: zazd, 5UF1)
**Compatibility:** UBS approximates Gaussian behavior; renders through standard 3DGS rasterizer. **Interpretability:** Temporal Beta parameter correlation with motion on D-NeRF scenes (Spearman ρ = 0.48-0.80) validates that learned parameters provide semantically meaningful representations without supervision.

---

## Common Themes Across Reviews

**Novelty and Technical Contributions:** UBS introduces three fundamental innovations—(1) Unified N-D anisotropic Beta primitives with per-dimension shape control, (2) Spatial-orthogonal Cholesky parameterization, (3) Beta-modulated conditional slicing with product-form opacity gate. These enable 41-73% parameter reduction and up to +8.27 dB PSNR improvement.

**Experimental Completeness:** We added more SOTA comparisons in Appendix A, efficiency metrics (Tables 8-9) in Appendix B, and per-primitive parameter analysis (Table 7).

**Theoretical Foundation:** Appendix section C and D provides Spatial-Orthogonal Cholesky derivation, rotation strategy validation (Table 10), Gaussian limitation analysis, and Beta kernel properties.

---

## Summary of Core Contributions

UBS makes substantial contributions:

(1) **Unified N-D Anisotropic Beta kernel framework** with per-dimension shape control;

(2) **Novel innovations** (Spatial-orthogonal Cholesky, Beta-modulated conditional slicing, and Beta-gated product-form opacity);

(3) **SOTA results** (highest PSNR on 3 benchmarks, +8.27 dB improvement);

(4) **Superior efficiency** (31-69% faster training, 43-47% memory reduction, 41-73% parameter reduction);

(5) **Comprehensive validation** (SOTA method comparisons, demonstrated stability). These establish Beta kernels as practical universal primitives.

---

We believe these revisions address all concerns and demonstrate that UBS represents a significant advance in radiance field rendering. We thank all reviewers for their feedback and welcome any additional suggestions during the discussion period.

---

### Author Response · Authors · 2025-12-01
**Summarized Comments to Area Chairs**

Dear Area Chairs,

We sincerely appreciate the reviewers’ time, thoughtful evaluations, and constructive suggestions. Their feedback has greatly helped us strengthen the clarity, analysis, and empirical validation of the paper. During the rebuttal period, we worked carefully to address every point raised through substantial revisions and additional experiments.

Since reviewers can no longer respond, we would like to briefly summarize the improvements made and explain why we now believe the paper merits positive consideration.

# Our Contributions
Universal Beta Splatting (UBS) introduces a unified N-dimensional anisotropic Beta kernel that jointly models spatial, angular, and temporal dimensions within a single primitive. Our approach is conceptually distinct from existing Gaussian-based methods:

1. Unified Primitive for Geometry, View Dependence, and Dynamics

UBS replaces several independent components used in prior work (e.g., 3DGS for geometry, SH for view dependence, deformation fields for dynamics) with a single primitive family capable of handling all three.
This unification offers:

- Joint control over spatial, angular, and temporal shape
- A single parameterization that smoothly expresses diffuse ↔ specular and static ↔ dynamic behavior
- Backward compatibility (Gaussians appear as a special case)

2. Novel Parameterization Enabling N-D Representation

- **Spatial-Orthogonal Cholesky Parameterization:**
   - Ensures PSD covariance, interpretable axes, cross-dimensional correlations, and efficient scalability to arbitrary N-D.

- **Beta-Modulated Conditional Slicing with Product-Form Opacity:**
   - Allows mean, covariance, and opacity to vary with viewing direction or time, providing an efficient alternative to SH encodings.
   - Enables a continuous decomposition of spatial vs. angular vs. temporal components, supporting the interpretability analyses in the paper.

3. Strong and Consistent Empirical Results

Across static, view-dependent, and dynamic benchmarks, UBS achieves:

- State-of-the-art PSNR on Mip-NeRF360, NeRF Synthetic, and the N3V real-world dynamic dataset
- 41–73% reduction in parameters
- 31–69% faster training
- Real-time rendering across all tasks

# Our Revisions
Below is a summary of our revision efforts. A more comprehensive description appears in our [Overall Response to All Reviewers](https://openreview.net/forum?id=51JEkjP0gF&noteId=FcgO9AqM00), and detailed per-question responses are provided in each reviewer thread.

- Fully rewritten, more self-contained preliminaries
- Expanded Related Work with additional state-of-the-art comparisons
- Detailed efficiency tables (FPS, memory usage, primitive counts)
- Added stability and convergence analyses
- Complete theoretical derivation of Eq. 8 (Spatial-orthogonal Cholesky parameterization)

# Reviewers’ Re-Evaluations After Rebuttal
Reviewer **4uqj**, who carefully reviewed the detailed responses and new experiments, explicitly noted that:

- All concerns were fully addressed
- The additional experiments and comparisons were sufficient and convincing
- The method’s novelty was clarified
- The score would be raised from **4 → 8 (Accept)**
- The reviewer was unable to update the score in the system due to a bug

The discussion unfortunately ended earlier than expected, and the other reviewers did not have time to provide updated feedback. However, because the clarifications and new experiments directly address their earlier concerns, we believe **their scores would likely have increased as well**.

- Reviewer 5UF1’s requests (theoretical justification, ablations, stability analysis, quantitative decomposition validation) were thoroughly resolved.
- Reviewer zazd’s concerns (backward compatibility, derivation clarity, stability analysis) were directly addressed.
- Reviewer bWeR’s requests (missing references, SOTA comparisons, primitive-count/FPS tables, self-contained exposition) were also fully incorporated.

Note that the same improvements that convinced reviewer 4uqj are relevant to the concerns raised by the other reviewers.

In summary, UBS provides a unified and empirically validated framework for radiance-field splatting, extending the expressive power of Gaussian Splatting while preserving efficiency and interpretability.
The extensive revisions made during the rebuttal address every point raised in the initial reviews, and the one reviewer who revisited the updated materials already increased their score from **4 → 8**.

We are grateful for the reviewers’ insights and suggestions, and we appreciate your time and consideration of our work.

Best Regards,

Authors

---

### Meta-Review · Area_Chair_CVun · 2026-01-07

**Summary:**

This paper recieves an initial rating of 2x marginal accepts, 1x accept and 1x marginal reject. The reviews are largely positive with mentions of strengths that include: 1) technically solid and conceptually original generalization of Gaussian Splatting via N-dimensional anisotropic Beta kernels. 2) combination of beta splatting with ND-GS is worth exploring; 3) proposed UBS framework can simultaneously model diverse scene types; 4) Extensive experiments across multiple benchmarks demonstrate UBS’s superior performance and strong generalization. 5) UBS features a pluggable architecture. Several weaknesses pointed out by the reviewers are: 1) novelty relative to prior high-dimensional splatting methods (e.g., 6DGS, DBS) is somewhat incremental ; 2) lack discussion on training stability and computational overhead; 3) paper simply applies beta control to 6D- and 7D-GS to achieve improved results, building on the strengths of these prior methods rather than introducing significant novelty.; 4)  fails to include many state-of-the-art methods in comparisons; 5) missing reference; 6) novelty concerns. However, the novelty concern mentioned by the reviewer who gave a marginal reject has been addressed in the rebuttal and discussion phases. The reviewer mentioned a raise of the score to accept. The AC follows the suggestions of the reviewers to accept the paper.

**Reviewer Concerns:**

Several weaknesses pointed out by the reviewers are: 1) novelty relative to prior high-dimensional splatting methods (e.g., 6DGS, DBS) is somewhat incremental ; 2) lack discussion on training stability and computational overhead; 3) paper simply applies beta control to 6D- and 7D-GS to achieve improved results, building on the strengths of these prior methods rather than introducing significant novelty.; 4)  fails to include many state-of-the-art methods in comparisons; 5) missing reference; 6) novelty concerns. However, the novelty concern mentioned by the reviewer who gave a marginal reject has been addressed in the rebuttal and discussion phases.

**Reviewer Scores:**

The positive scores reviewers would maintain their positive ratings. The negative score reviewer has mentioned a raise of the score to accept.

---

### Decision · Program_Chairs · 2026-01-26

Accept (Poster)